# Effects of parental care on skin microbial community composition in poison frogs

Marie-Therese Fischer[1,2]*, Katherine S Xue[1], Elizabeth K Costello[3], Mai Dvorak[1], Gaelle Raboisson[1], Anna Robaczewska[3], Stephanie N Caty[1], David A Relman[3,4,5], Lauren A O'Connell[1,6]

[1]Department of Biology, Stanford University, Stanford, United States; [2]Division of Microbial Ecology, University of Vienna, Vienna, Austria; [3]Department of Medicine, Stanford University School of Medicine, Stanford, United States; [4]Department of Microbiology & Immunology, Stanford University School of Medicine, Stanford, United States; [5]Infectious Diseases Section, Veterans Affairs Palo Alto Health Care System, Palo Alto, United States; [6]Wu Tsai Institute for Neuroscience, Stanford University, Stanford, United States

*For correspondence:
fi5cher@stanford.edu

## eLife Assessment

This study provides an **important** perspective on the influence of parental care in the establishment of the amphibian microbiome. Through a combination of cross-fostering experimental work, comparative analysis, and developmental time series, the authors provide **compelling** evidence that vertical transmission through care is possible, and **solid** but somewhat preliminary evidence that it plays a significant role in shaping frog skin microbiomes in nature or across time. This work will be of interest to researchers studying the evolution of parental care and microbiomes in vertebrates.

**Abstract** Parent-offspring interactions constitute the first contact of many newborns with their environment, priming community assembly of microbes through priority effects and shaping host health and disease. Microbe acquisition during parental care is well studied in humans and agriculturally relevant species but remains poorly understood in other vertebrate groups, such as amphibians. Here, we investigate vertical transmission of skin microbiota in poison frogs (*Dendrobatidae*), where fathers transport tadpoles piggyback-style from terrestrial clutches to aquatic nurseries. We found that substantial bacterial colonization of embryos begins after hatching, suggesting that the vitelline envelope acts as a microbial barrier. A cross-foster experiment demonstrated that poison frogs performing tadpole transport serve as a source of skin microbes for tadpoles on their back. To study how transport impacts skin communities of tadpoles in an ecologically relevant setting, we sampled sympatric species that do or do not exhibit tadpole transport in their natural habitat. We did not find a higher degree of similarity between microbial communities of tadpoles and adults in species that transport their offspring compared to those that do not. Similarly, communities of tadpoles were no more similar to their caregiver than to unrelated adults, indicating that most caregiver-associated microbes do not remain in tadpole communities long-term. Nonetheless, some taxa persisted on tadpoles over development. This study is the first to demonstrate that offspring transport facilitates transmission of parental skin microbes in anurans.

**eLife digest** The first microbes that animals encounter after birth can impact their health and development for life. In many species, these microbes are passed from parent to offspring during close contact or while caring for the young – a process known as vertical transmission. While this has been well studied in mammals, we know much less about how, or even if, microbes are passed from parent to offspring in amphibians.

Amphibians rely on healthy skin to breathe and regulate water and salt levels, and skin-dwelling microbes also help protect them from infections. Many tropical frogs also display striking forms of parental care. For example, many poison frog parents carry their newly hatched tadpoles from land to small pools of water on their backs. This journey can take hours or even days, creating a perfect opportunity for microbes to transfer from parent to offspring through skin-to-skin contact.

Despite this, it remains unclear how parental care in frogs influences the composition of microbes in their offspring. To find out more, Fischer et al. explored when and how poison frog tadpoles acquire their skin microbes. First, the researchers examined whether frog embryos are exposed to microbes while still developing inside the egg. Then they tested whether physical contact during transport – when the parent carries the tadpole – actually passes microbes to the tadpole, and whether those microbes remain on the tadpole's skin as it grows.

To answer these questions, Fischer et al. combined lab experiments with field research. Using DNA sequencing to study the bacteria living on tadpole skin, Fischer et al. found that most microbes begin colonizing only after the tadpoles hatch from their egg membranes, just before they are carried to water by a parent. In laboratory experiments, tadpoles picked up microbes through direct contact with the skin of the parent carrying them. In wild frogs, it was found that while microbes do transfer, only a few persisted on the tadpoles' skin over time.

These findings are important for understanding how microbial communities are formed and how parental care shapes this process. For frogs, skin microbes provide a crucial defense against deadly diseases such as chytrid fungus. By understanding how and when amphibians acquire beneficial microbes, we can help protect the habitats and behaviors that support this process and better conserve vulnerable species.

## Introduction

Host-associated microbial communities contribute to health and facilitate ecological adaptations by playing critical roles in host development, nutrition, pathogen exclusion, and immune response (*Belkaid and Hand, 2014*; *Donald and Finlay, 2023*; *Gensollen et al., 2016*; *Mazmanian et al., 2005*; *Reynolds and Bettini, 2023*; *Sampson and Mazmanian, 2015*; *von Frieling et al., 2018*). Across host species, early exposure to microbes can exert lasting influences on the assembly and functionality of the host's microbiota (*Al Nabhani and Eberl, 2020*; *Powell et al., 2014*; *Warne et al., 2017*; *Zepeda Mendoza et al., 2018*). Timing of microbial contact is particularly critical in newborns and can prime community assembly through priority effects, where species that arrive early impact the establishment of later colonizers (*Debray et al., 2022*; *Fukami, 2015*; *Sprockett et al., 2018*). In many species, the transmission of host-adapted microbes is facilitated by parental care strategies, such as feeding, grooming, and direct skin contact (for review see e.g. *Wang et al., 2020* for humans, *Klug and Bonsall, 2014* for animals; see also *Blyton et al., 2022*; *Chen and Garud, 2022*; *Pascoe et al., 2017*; *Sprockett et al., 2020*; *Sylvain and Derome, 2017*). To date, most studies investigating effects of exposure to microbiota during parental care center on humans and livestock, while vertical transmission remains largely unexplored in other classes of vertebrates (but see *Kouete et al., 2023*). Therefore, investigating early-life microbial colonization in other species that have dedicated parenting strategies offers opportunities to understand microbe-host interactions in a wider variety of ecologically relevant contexts.

Acquisition of host-adapted skin microbes is especially important in amphibians, which rely on healthy skin for physiological processes such as respiration, osmoregulation, immune response, and barrier function (*Akat Çömden et al., 2023*; *Duellman and Trueb, 1986*; *Varga et al., 2019*). The amphibian skin microbiota also serves to exclude pathogens, as the skin microbiome can confer resistance to infections by the deadly chytrid fungus *Batrachochytrium* (*Alexiev et al., 2023*; *Harris et al.,*

*2006*; *Harris et al., 2009*; *Loudon et al., 2014*; *Vredenburg et al., 2011*; *Woodhams et al., 2007*), which decimates amphibian populations worldwide (*Daszak et al., 2003*; *Luedtke et al., 2023*; *Wake and Vredenburg, 2008*). Many studies have characterized microbe-host interactions in adult frogs in the context of chytrid infections, but few have focused on microbiomes associated with tadpoles (*Fontaine et al., 2022*; *Santos et al., 2023*; *Warne et al., 2017*; *Warne et al., 2019*; *Weinfurther et al., 2023*). Those studies that have looked at tadpoles have primarily characterized the changes in bacterial communities across development and metamorphosis (discussed e.g. in *Hughey et al., 2017*; *Kueneman et al., 2014*; *Prest et al., 2018*; *Warne et al., 2017*; *Warne et al., 2019*). Furthermore, most of this research has focused on temperate-region, pond-spawning species, where opportunities to investigate effects of parental care on community assembly are limited. These species generally lay many eggs at once into bodies of water to avoid desiccation and larvae develop without further parental contact or care. In the tropics, a warm and humid climate has favored the progression from aquatic to terrestrial reproduction, where clutches are laid on land and parents then care for their offspring (*Hanken, 1999*; *Wells, 2007*). Neotropical frogs show more diverse reproductive strategies where parents construct foam nests, attend clutches, defend eggs against predators, and guard, transport, or feed their tadpoles (reviewed e.g. in *Schulte et al., 2020*, see also *Crump, 2015*; *Delia et al., 2013*; *Requena et al., 2009*; *Warkentin, 1995*). Priority effects influence community assembly of frog embryos (*Jones et al., 2024*; *Jones et al., 2023*) but previous studies have found no direct evidence for vertical transmission of microbes during egg attendance (*Hughey et al., 2017*). However, this form of offspring care does not involve direct contact between the parents and offspring as developing tadpoles in the clutch are surrounded by several layers of gelatinous jelly and a double vitelline envelope before they hatch (see *Altig and McDiarmid, 2007* and *Méndez-Tepepa et al., 2023* for a review). While the vitelline layer may serve as an antimicrobial barrier in chickens (*Guyot et al., 2016*; *Mann, 2008*), studies of its role in microbial colonization of anuran embryos are currently lacking.

Parental care involving skin-to-skin contact has developed multiple times independently in Neotropical dendrobatoid frogs (Superfamily *Dendrobatoidea*), commonly referred to as poison frogs (*Grant et al., 2017*; *Weygoldt, 1987*). Closely related species in this clade display a variety of parenting behaviors ranging from egg attendance and transport of hatched tadpoles to provisioning larvae with unfertilized eggs (*Crump, 1974*; *Ringler et al., 2023*; *Wells, 2007*). The most widespread form of care beyond clutch attendance is tadpole transport, where parents shuttle hatched tadpoles piggy-back style from terrestrial clutches to water pools (*Furness and Capellini, 2019*). Transport can last multiple hours to days, where tadpoles adhere tightly to the backs of their caregivers (*Pašukonis et al., 2019*; *Pröhl and Berke, 2001*; *Ringler et al., 2013*, p. 201; *Wells, 1980*). Therefore, poison frogs provide the opportunity to evaluate the effects of parental care on offspring skin microbial community assembly. The adaptive value of parental care in this clade has received considerable attention (reviewed in *Schulte et al., 2020*; *Summers and Tumulty, 2014*), but how reproductive strategies involving skin-to-skin contact impact microbial colonization of tadpoles and whether these effects persist to later developmental stages has, to our knowledge, not been studied in any anuran to date.

The variable poison frog *Ranitomeya variabilis (Rv)* (Zimmermann & Zimmermann, 1988) is particularly well-suited to study vertical transmission of microbes during tadpole transport. This small species is diurnal, commonly found across South America, and readily reproduces in captivity. Males of this species often assist their larvae with hatching (*Brown et al., 2008*) and carry tadpoles on their back for up to 48 hr (*Lötters et al., 2007*). The cannibalistic tadpoles grow up in individual pools where they feed on algae and leaf debris until they complete their development after about five months. In the natural study population located in the reserve 'Les Nouragues' in French Guiana, these frogs live on rocky outcrops, use bromeliads as a resource for reproduction and shuttle tadpoles from the oviposition site in arboreal plants to water reservoirs in leaf axils of terrestrial bromeliads (*Poelman et al., 2013*; *Poelman and Dicke, 2007*; *Poelman and Dicke, 2008*; *Sarthou, 2001*). This species is sympatric with other anuran species with differing parenting strategies: the poison frog *Allobates femoralis (Af)* shuttles tadpoles to water but transporting periods are shorter and tadpoles live in groups. The leptodactylid frog *Leptodactylus longirostris (Ll)* deposits eggs in ephemeral rock pools and does not exhibit parental care. These cohabiting species are potentially useful for determining if reproductive strategies impact patterns of microbial diversity of tadpoles.

In this study, we combine laboratory experiments with sampling of anuran populations in the wild to examine vertical transmission of microbes during tadpole transport and address implications for

community structure across life stages. In the laboratory, we investigated microbial colonization of hatched and unhatched *Rv* embryos and then tested the hypothesis that transporting frogs serve as a source for bacteria on tadpole skins using cross-foster experiments. We complemented these experiments with a comparative field study examining microbiomes associated with tadpoles and adults of three anuran species with varying parenting behaviors to examine how shuttling affects community composition on the skin of tadpoles in their native habitat. To our knowledge, this is the first study to document and characterize vertical transmission of microbes during parental care in any anuran.

## Results

### Microbial colonization of a poison frog embryo occurs after hatching

Clarifying the timing of initial microbial colonization is essential for determining whether and how priority effects mediate community assembly of vertically transmitted microbes in early life, or whether these microbes arrive into an already established microbial landscape. We used non-sterile frogs of our captive laboratory colony to determine the timing of microbial colonization of poison frog embryos. Embryos within the vitelline membrane exhibit a characteristic C-shaped posture due to the membrane's restriction on their movement and adopt a straight posture after hatching (*Figure 1—figure supplement 1*). First, we qualitatively examined bacterial presence in *Rv* egg jellies and embryos across tadpole developmental stages using a broad-range PCR targeting the 16 S rRNA gene. We manually separated embryos in the vitelline envelope from the surrounding jelly and processed each for DNA extraction (*Figure 1A*). Bacteria were detected in egg jellies across all developmental stages (formation of the dorsal lip to hatchling) but were not detected in embryos prior to hatching from the vitelline membrane (N=12) (*Figure 1—figure supplement 1B*). After hatching, bacteria were detected in tadpoles that remained in the jelly and in tadpoles that were transported by their caregiver (*Figure 1—figure supplement 1B*). Next, we confirmed differences in bacterial load between jelly and embryo by examining variations in 16 S rRNA gene copy numbers using a droplet digital PCR (ddPCR) approach suited for low biomass samples (*Abellan-Schneyder et al., 2021*). We found that unhatched embryos contained on average 2942 copies of the 16 S rRNA gene per µl (min = 742, max = 11584, std = ± 3086) while jellies contained on average 6128 times as many (min = 457417, max = 66115451, mean = 18029604 ± 18586471; Kruskal-Wallis (KW): chi-squared=18.6667, df = 2, p<0.00001) (*Figure 1B*, *Figure 1—figure supplement 1C*, *Supplementary file 1a*). 16 S rRNA gene copy concentrations were on average 8310 times higher in hatched relative to unhatched embryos (min = 737340, max = 48159335, mean = 24448337 ± 33532414; KW: p=0.0195), but did not differ significantly from concentrations detected in the jelly (KW: p=0.5) (*Figure 1—figure supplement 1C*, *Supplementary file 1a*).

### Evidence for vertical transmission of microbiome during parental care in laboratory-reared poison frogs

We used a cross-foster design to test the hypothesis that frogs carrying tadpoles on their backs serve as a source of microbes for communities on tadpole skin. Siblings of the same clutch (N=6 clutches from 2 *Rv* pairs) were randomly assigned to three groups: (1) not carried (N=6 tadpoles), (2) carried on the back of their biological parent for 6 hr (N=9 tadpoles), or (3) carried by a heterospecific poison frog (*Oophaga sylvatica, Os*) for 6 hr (N=9 tadpoles) (*Figure 1C*). For this design, sequence-based surveys of amplified 16 S rRNA genes were used to assess the composition of skin-associated microbial communities from tadpoles and their adult caregivers (i.e. the frogs carrying the tadpoles, typically referred to as 'transporting' frogs). We found that bacterial community composition on caregiver skin was primarily shaped by host species (*Rv* versus *Os*; Adonis: Bray-Curtis: $F_{1, 12}$ = 5.6538, $R^2$=0.33949, p=0.001; unweighted Unifrac: $F_{1, 12}$ = 2.0078, $R^2$=0.15435, p=0.013), allowing us to distinguish between these potential sources of tadpole-colonizing microbes (*Figure 1—figure supplement 2A*). Indeed, we found that after 6 hr of being carried on the back, bacterial community composition on *Rv* tadpole skin was influenced by caregiving species (*Rv* versus *Os*; Adonis: Bray-Curtis: $F_{1, 12}$ = 1.75, $R^2$=0.099, p=0.017; Unifrac: $F_{1, 12}$ = 1.67, $R^2$=0.095, p=0.015). Despite their distinct compositions, we observed substantial overlap between communities of tadpoles carried by *Os* and *Rv* (*Figure 1—figure supplement 2B*), possibly reflecting similarities in tadpoles' skin communities arising from a shared clutch environment after hatching until transport.

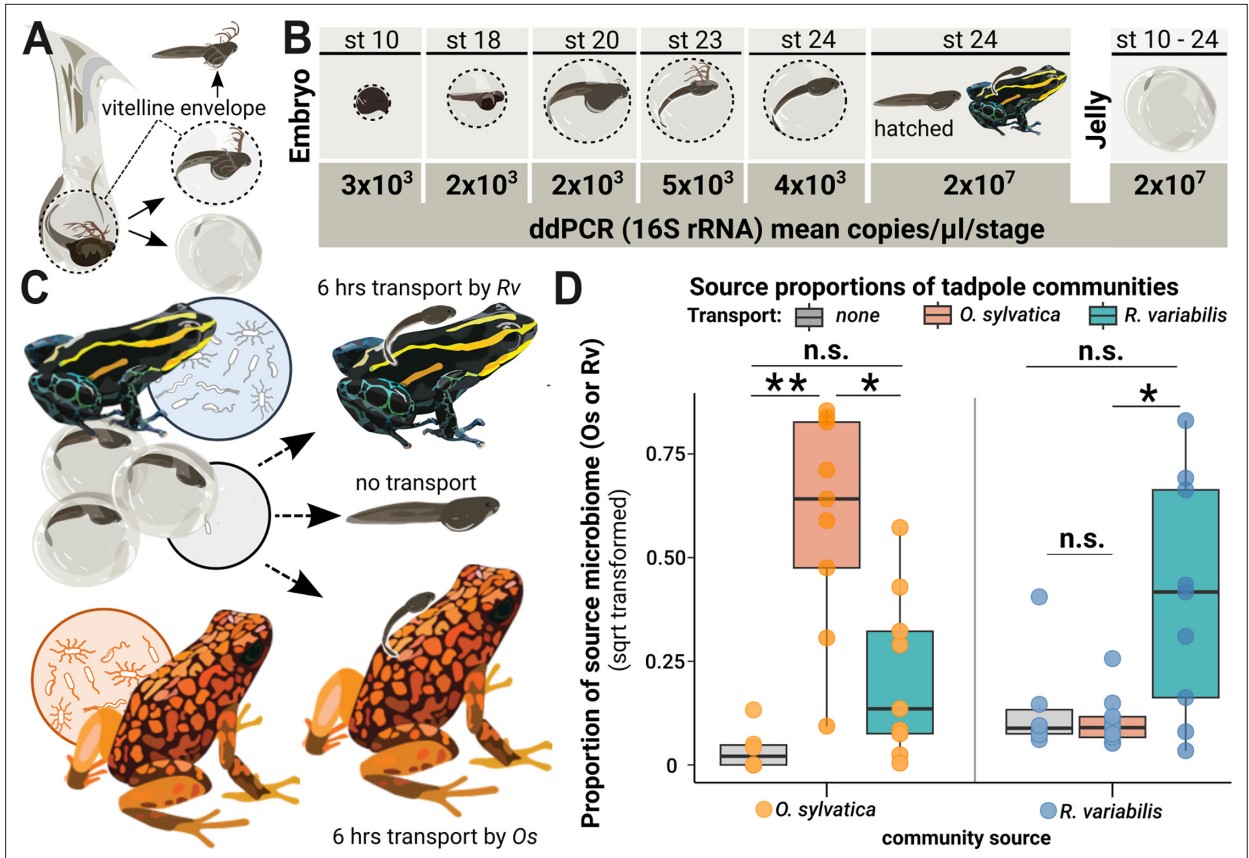

**Figure 1.** Microbes colonize a poison frog embryo after hatching and are vertically transmitted during tadpole transport. (**A**) Embryos in the vitelline envelope (dashed line) were manually separated from jelly using tweezers. The vitelline envelope (dashed line) containing the embryo was transferred to sterile water and opened to free the embryo. The embryo was washed in fresh sterile water before homogenization. (**B**) 16 S rRNA gene copy number variations across developmental stages of embryos and jelly. DNA was isolated from whole embryos (N=14) and jellies (N=12) of different developmental stages and tested for bacterial presence using droplet digital PCR (ddPCR). Rounded mean copy numbers/µl/stage are displayed. (**C**) After hatching, siblings of a clutch were either (1) not transported (middle arrow, N=6), (2) transported by their biological parent (upper arrow, N=7), or (3) transported by a foster poison frog of a different species (*Oophaga sylvatica*) (lower arrow, N=9). (**D**) We performed 16 S rRNA gene-specific amplicon sequencing targeting the V4 region on swabs from the transporting frogs and the skins of the transported tadpoles and used Sourcetracker to identify the sources of taxa that had been acquired by tadpoles. The function was trained on communities of adult *Ranitomeya variabilis* (*Rv*) and *Oophaga sylvatica* (*Os*) that had served as caregivers. Source proportions of both species (*Os*: orange dots and *Rv*: blue dots) were determined for each tadpole (N=24), resulting in two data points per tadpole. Source proportions (*Os*: orange dots, *Rv*: blue dots) were determined in tadpoles transported by *Os* (orange boxes), *Rv* (blue boxes), and non-transported tadpoles (gray boxes) and compared with a Kruskal-Wallis test. P-values were adjusted for multiple comparisons (Benjamini-Hochberg). Boxplots show the median and interquartile range (IQR); error bars represent values within 1.5× the IQR, significances are indicated by * (<0.05), and ** (<0.01).

The online version of this article includes the following source data and figure supplement(s) for figure 1:

**Figure supplement 1.** Microbes colonize a poison frog embryo after hatching from the vitelline membrane.

**Figure supplement 1—source data 1.** Original gel pictures containing used to create *Figure 1—figure supplement 1B*, indicating the relevant bands and sample names.

**Figure supplement 1—source data 2.** Original, unedited pictures used to create *Figure 1—figure supplement 1B*.

**Figure supplement 2.** Transporting frogs serve as a source of microbes for transported tadpoles.

We used 'ourcetracker' (*Knights et al., 2011*) to determine the relative contribution of the transporting frogs as a source of bacteria detected in tadpoles' skin after 6 hr of transport. Tadpoles of a clutch that were transported by *Os* always shared a higher proportion of their communities with this heterospecific caregiver than siblings that were not transported (KW: chi²=14.4696, df = 2, p=0.0003; difference in proportion: median = 39.33, min = 0.872%, max = 68.478%) or siblings transported by the biological parent (KW: p=0.015; percentage of higher proportion: median = 37.07%, min = 0.874 %, max = 72.97%) (*Figure 1D*, *Supplementary file 1b*). Similarly, all but two

transported tadpoles shared a larger proportion of their communities with the transporting species than with the non-transporting species (p=0.001) (*Supplementary file 1b*). Transported tadpoles (N=18) generally shared higher proportions of their skin communities with the transporting species than did non-transported tadpoles (N=6) (Kruskal-Wallis chi$^2$=19.4029, df = 2, p=0.0001) (*Figure 1— figure supplement 2D*). Non-transported tadpoles were colonized by between 5 and 57 Amplicon Sequence Variants (ASVs) (median = 12). We conducted a separate analysis defining non-transported tadpoles as potential microbe sources to represent community proportions that were acquired in the clutch before caregiver contact (*Figure 1—figure supplement 2C*). We found that transported tadpoles shared between 0 and 46.99% (median = 13%) of their communities with non-transported tadpoles. Sourcetracker detected higher proportions of clutch-acquired microbes in tadpoles transported by *Os* (pairwise Wilcoxon test, p=0.02). This likely reflects the higher degree of similarity between the source clutch and *Rv*, as the clutch remained in the *Rv* tank during development, making it harder to distinguish between these sources. Overall, our results suggest that microbes are vertically transmitted during tadpole transport in a poison frog.

## Tadpoles of the poison frogs *Ranitomeya variabilis* and *Allobates femoralis* host more diverse skin communities than tadpoles of the leptodactylid frog *Leptodactylus longirostris*

To investigate how vertical transmission of microbes during tadpole transport affects microbial skin communities of tadpoles in an ecologically relevant context, we studied the skin microbiome composition and diversity of *Rv* tadpoles and adults in a natural population. To assess the broader relevance of our findings to frog species that transport their tadpoles, we also sampled two sympatric species with differing reproductive strategies: *Allobates femoralis (Af)* and *Leptodactylus longirostris (Ll)* (*Figure 2A*). In contrast to *Rv* that transports its cannibalistic tadpoles to individual pools in bromeliads, *Af* lives in forest leaf litter and shuttles multiple non-cannibalistic tadpoles to the same pool where they grow up gregariously. *Ll* is a nocturnal leptodactylid frog inhabiting the rock savanna that does not exhibit tadpole transport but deposits eggs directly in ephemeral rock pools where the larvae grow up together (*Figure 2—figure supplement 1*).

We compared bacterial communities from 137 skin swabs of adult frogs, tadpoles, and the aquatic environment belonging to the tadpoles of these three species. We sampled 44 adults and 21 tadpoles (Gosner stages 29–41, categorized as 'medium' and 'large') of the species *Rv*, 10 adults and 14 tadpoles (Gosner stages 34–41, categorized as 'medium' and 'large') of the species *Ll*, and 10 adults and eight tadpoles (Gosner stages 25–26, categorized as 'small') of the species *Af* (*Supplementary file 1c*). Additionally, we monitored the growth of 184 *Rv* tadpoles in separate bromeliad pools for up to 82 days. Out of 45 tadpoles that were monitored for over 50 days, 13 (three 'small' and 10 'medium') tadpoles did not advance in their sizing category, reflecting that the complete larval development in this population spans multiple months. Cannibalistic *Rv* tadpoles grow up in isolated pools that were sampled as single replicates when collecting the tadpoles. Due to their development in groups, *Af* and *Ll* tadpoles were collected from fewer water bodies and represent a narrower developmental window (stages 34–41 for *Ll* and stages 25 and 26 for *Af*). If multiple tadpoles were collected from the same aquatic environment (*Af* and *Ll* tadpoles), we sampled triplicates of the respective environment.

For each species, the aquatic environment of the tadpoles displayed a higher average phylum-level diversity than the inhabiting tadpoles or adults (*Supplementary file 1c*). As previously reported for Neotropical frogs (*Hughey et al., 2017*), ASVs belonging to Phylum Proteobacteria dominated the microbial community across all species and life stages (mean relative abundance: 70.9%), followed by Bacteroidota (10.3%), Actinobacteriota (4.1%), Firmicutes (4%), Verrucomicrobiota (2.3%), and Planctomycetota (1.1%) (*Figure 2—figure supplement 2*). The abundance of the frog pathogen *Batrachochytrium dendrobatidis (Bd)* in the rock savanna was low, with detection in two samples of *Rv* adults and one *Rv* tadpole, but no positive *Ll* sample. We detected *Bd* on one adult *Af* individual collected in primary forest but not on tadpoles of this species collected in the same area. In all species, tadpole communities contained fewer taxa with known *Batrachochytrium*-growth inhibiting function than did environmental samples or adults (*Supplementary file 1c*).

We found no significant differences in the observed ASV richness, diversity (Shannon), or evenness between tadpoles and adults of the poison frogs *Rv* and *Af*. In contrast, *Ll* tadpoles were significantly less diverse than *Ll* adults (*Figure 2B*; *Supplementary file 1d*). We followed up on this finding by

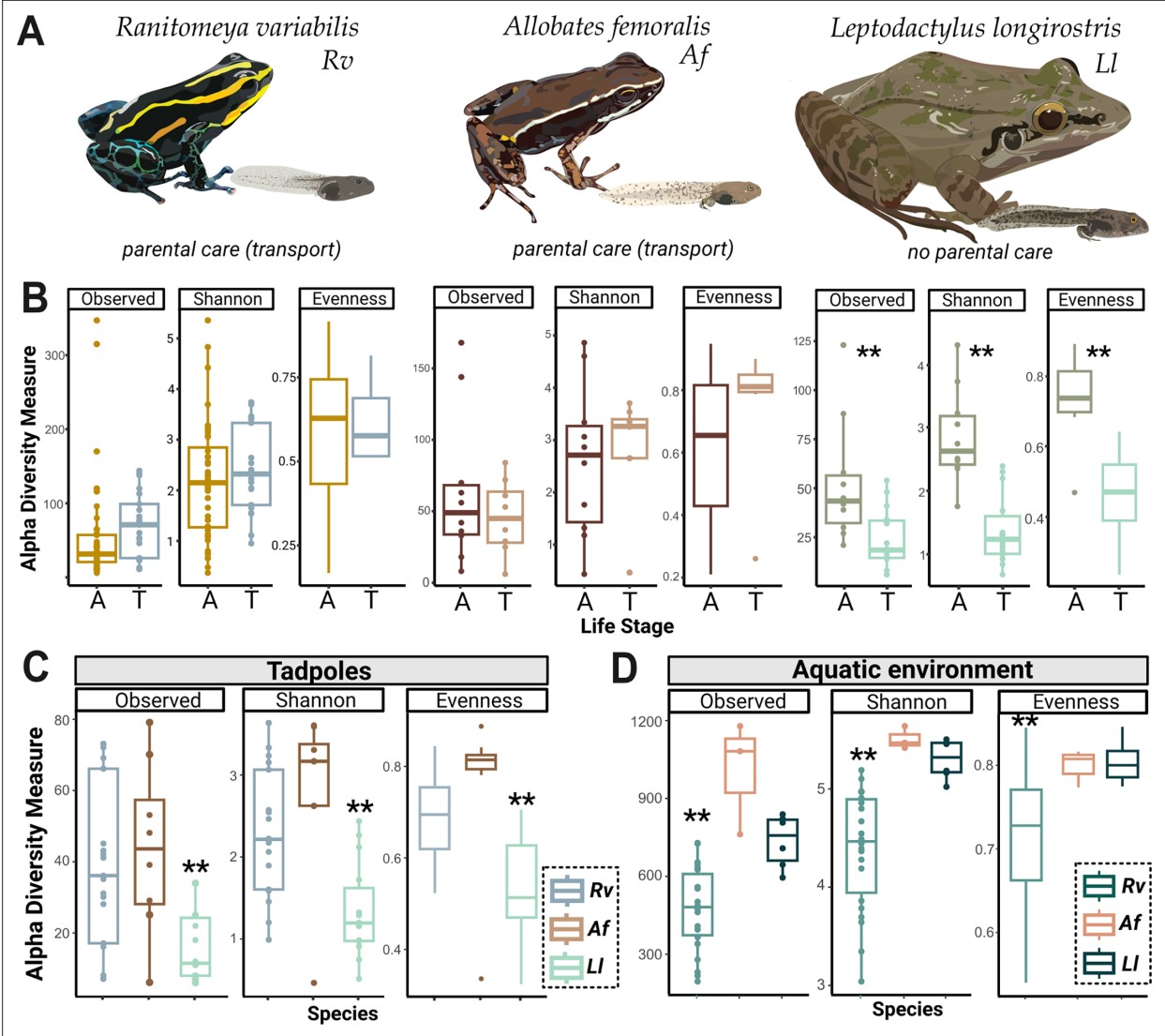

**Figure 2.** Tadpoles skin microbiome is shaped by their environment and is more diverse in *Ranitomeya variabilis* (*Rv*) and *Allobates femoralis* (*Af*) compared to *Leptodactylus longirostris* (*Ll*) tadpoles. (**A**) We compared the skin microbiome of three anuran species: two species of poison frogs inhabiting different habitats that transport their offspring (*Rv* and *Af*) and a leptodactylid frog (*Ll*) that deposits its eggs in water without transporting the tadpoles. N (*Rv*)=44 adults and 21 tadpoles, N (*Ll*)=10 adults and 14 tadpoles, N (*Af*)=10 adults and ight tadpoles. (**B**) Alpha diversity measures (observed Amplicon Sequence Variant ASV richness, Shannon diversity, and evenness) for tadpoles (T) and adults (A) of each species were compared. Differences were determined with an ANOVA or Kruskal-Wallis test, p-values were adjusted for multiple comparisons (Benjamini-Hochberg). (**C**) Comparison of ASV richness, Shannon diversity, and evenness of communities associated with poison frog tadpoles (*Af* or *Rv*) and non-poison frog species (*Ll*). (**D**) Comparison of Shannon diversity and evenness of communities associated with the aquatic habitats of *Af*, *Rv*, and *Ll*. Bars in boxplots represent median values, error bars represent values within 1.5× the interquartile range (IQR), significant p-values <0.01 are indicated by **. The dataset was separately rarefied to the lowest read depth of each comparison.

The online version of this article includes the following figure supplement(s) for figure 2:

**Figure supplement 1.** Study species and their habitat.

**Figure supplement 2.** Phyla distribution per species and group.

comparing the diversity of tadpole-associated communities across species and found higher microbial diversity and evenness in *Rv* and *Af* tadpoles compared to *Ll* tadpoles (*Figure 2C*). Differences in community diversity of tadpoles were not a reflection of habitat diversity: microbial communities associated with *Rv* aquatic environments were less diverse and even compared to those in *Af* and *Ll* habitats (*Figure 2D*). These variations may, therefore, reflect differences in life history traits among the three species.

## Tadpole transport is not associated with a higher degree of similarity between adult and tadpole skin microbiotas

To assess how microbes transmitted during parental care shape communities associated with tadpoles, we tested the hypothesis that tadpoles of the transporting species *Rv* and *Af* would have communities more similar to conspecific adults compared to tadpoles of the non-transporting species *Ll*. We first evaluated variation in bacterial communities associated with adults, tadpoles and their microenvironment by using a principal coordinate analysis (PCoA) to construct ordinations based on Bray-Curtis dissimilarities to explore factors that influence skin community composition (*Figure 3A*). Tadpoles of all species clustered separately from adults, each other, and their aquatic environment (PERMANOVA on genus level, df = 8, $R^2$=0.52909, *F*=18.117, p adj = 0.001) (*Figure 3A and B*; *Supplementary file 1e*). Consistent with findings in pond-spawning anurans (*Kueneman et al., 2014*; *Prest et al., 2018*), skin communities varied significantly across life stages in all species (PERMANOVA on genus level: species: df = 5, $R^2$=0.29674, *F*=11.139, p=0.001, life stage: df = 2, $R^2$=0.23148, *F*=20.331, p=0.001). Parental care also explained some of the observed variance (PERMANOVA on genus level: df = 2, $R^2$=0.07513, *F*=5.4831, p=0.001). To further examine patterns of similarities between life stages, we compared the core communities of adults, tadpoles, and their aquatic environments across a range of prevalence and abundance cutoffs prevalence 100% or 75%; relative abundance no cutoff, ≥0.1% (low) and ≥1% (high) (*Supplementary file 1f*). Independent of parenting strategy and species, tadpoles shared core communities of their microbiome with their environments, but not with adult individuals. Adult poison frogs never shared core taxa with the aquatic environment of their tadpoles (*Figure 3C*; *Supplementary file 1f*), even though adults were commonly found in the nurseries. In contrast, the aquatic environment contributed substantially to the skin community of adult *Ll*, which shared high- and low-abundance (≥0.1%) core genera with the pond water, but not with the pond-inhabiting tadpoles (*Figure 3C*; *Supplementary file 1g*).

To assess if microbes transmitted during tadpole transport constitute a stable part of skin communities associated with tadpoles, we compared community distances between skin communities of tadpoles and adults of the three frog species. We predicted that tadpoles and adults might have more similar skin microbiomes in the transporting species *Rv* and *Af* than in the non-transporting species *Ll*. We found no difference in community distances between tadpoles and adults in *Af* and *Ll* (iterated beta regression averaged over 999 random subsamples of similar sample size (IBR): Unifrac: p=0.3). Communities associated with *Rv* tadpoles were more dissimilar from adult communities than microbiota of *Ll* tadpoles and adults (IBR: Unifrac: p=0.008). Taken together, our results indicate that microbes acquired during tadpole transport do not create lasting similarities in composition between tadpoles and adults of wild populations.

## After 4 weeks, transported *Ranitomeya variabilis* tadpoles retain isolated members of their caretaker's core community

To test how offspring transport in *Rv* affected the community composition in tadpoles, we conducted a field experiment where we reared transported (N=10) and non-transported (N=10) tadpoles in artificial cups. Transported tadpoles were collected from the backs of their caregiver (N=8), while we bypassed parental care in the non-transported group by transferring eggs to cups before the tadpoles hatched (*Figure 4A*). Two out of ten non-transported tadpoles died before sampling. Richness and evenness of tadpole communities did not differ significantly between experimental conditions (ANOVA: Observed ASVs: df = 1, *F*=2.7234, p.adj=0.3552; Shannon: df = 1, *F*=1.2067, p.adj=0.8646; evenness: df = 1, *F*=0.0009, p.adj=2.9313), though PCoA on Unifrac distances revealed differences in microbiome composition attributable to the presence or absence of transport (*Figure 4B*) (PERMANOVA: df = 1, $R^2$=0.11759, *F*=2.1321, p=0.002). Compared to transported tadpoles, non-transported tadpoles showed less overlap of genera in their microbial community with their aquatic environment (*Figure 4C*). Specifically, the genera *Pelomonas*, *Rhodomicrobium* (Proteobacteria), *Mycobacterium* (Actinobacteriota), and *Candidatus Koribacter* (Acidobacteriota) primarily colonized transported tadpoles, while the genus *Limnohabitans* (Proteobacteria) was found on non-transported tadpoles. *Cetobacterium* and *Burkholderia-Caballeronia-Paraburkholderia* were found to colonize tadpoles from both experimental conditions, but at different abundances (ANCOMBC 2, p<0.05; *Supplementary file 2*).

We further tested the hypothesis that related tadpole-adult pairs with a history of direct contact would have more similar communities than unrelated tadpole-adult pairs without such a history. One

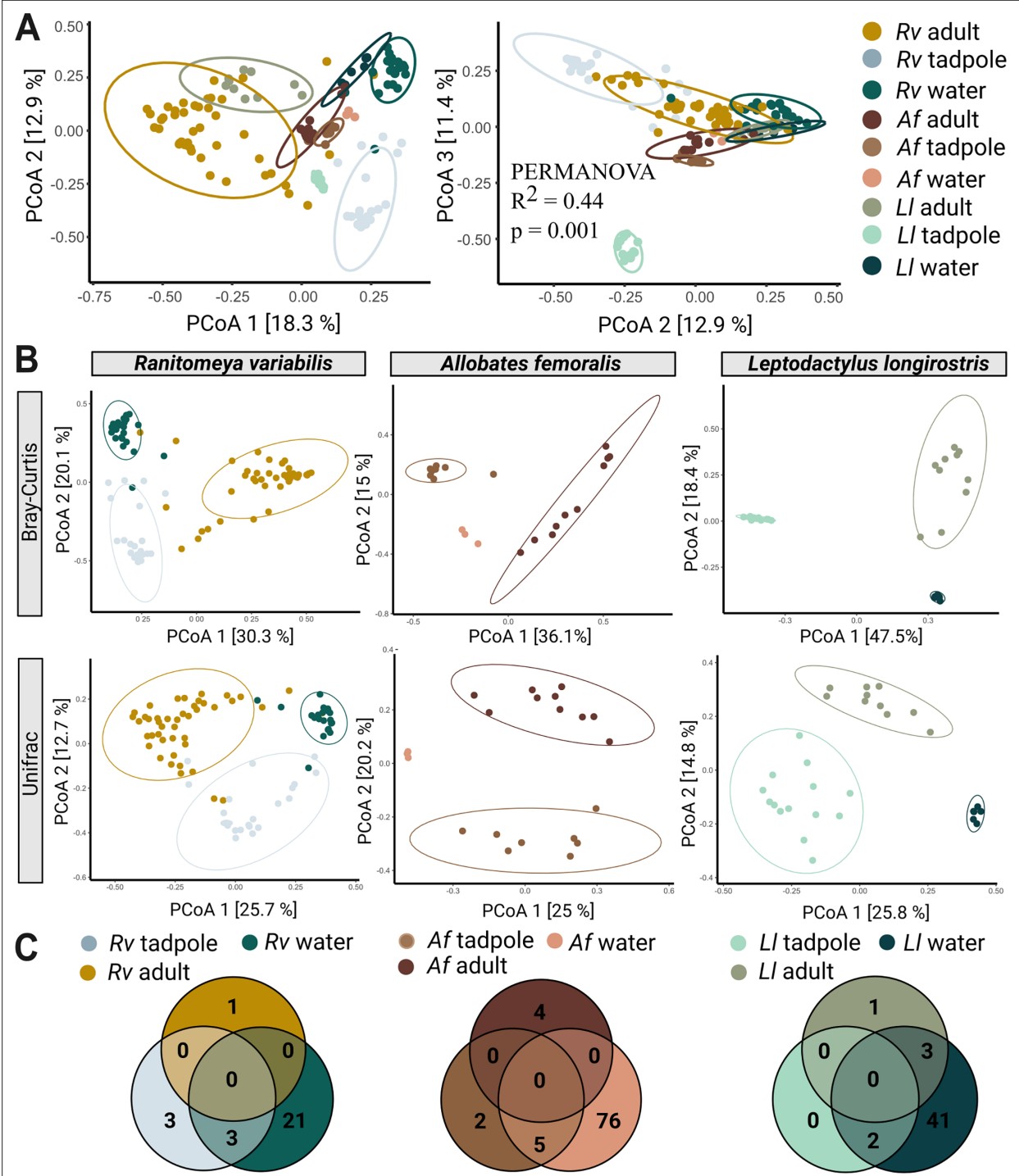

**Figure 3.** Species, life stage, and parental care affect clustering of microbial communities. (**A**) In a Principal Coordinate Analysis constructed with Bray-Curtis distances (axis 1 and 2 on the left, axes 2 and 3 on the right) tadpoles cluster significantly differently from each other, adults, and their aquatic environment. Significances were determined with a PERMANOVA followed by a pairwise Adonis post hoc test. Points in ordination plots represent the communities of each sample, circles represent confidence ellipses. N (*Ranitomeya variabilis*, *Rv*)=44 adults and 21 tadpoles, N (*Leptodactylus longirostris*, *Ll*)=10 adults and 14 tadpoles, N (*Allobates femoralis*, *Af*)=10 adults and eight tadpoles. (**B**) Principal Coordinate Analysis constructed with Bray-Curtis and Unifrac distances for adults, tadpoles, and aquatic environment of each species. (**C**) Number of core species (prevalence >75%, relative abundance >0.1%) shared between adults, tadpoles, and the respective aquatic environment of each species.

month after transport, we found that skin communities of tadpoles were not more similar to the transporting parent than to unrelated adults of the population (iterated beta regression averaged over 999 random subsamples of similar sample size (N=10) (IBR): Unifrac: p=0.27), confirming our results from the population sampling. However, a Sourcetracker analysis on unrarefied data revealed notable relative contributions of the parental skin community to the community on four transported tadpoles (tadpoles 1, 2, 6, 8) (min = 0% to max = 17%, median = 0%, IQR = 10.5; *Supplementary file 1h*). We followed up on this finding by directly identifying taxa shared between caregivers and the respective transported offspring without imposing an abundance cutoff. We found that 8 out of 10 tadpoles shared one or two ASVs with the frog that had transported them in relative abundances ranging from 0.02 to 19.3% (*Figure 4D*; *Supplementary file 1h*). Shared taxa belong to the genera *Akkermansia, Elsteraceae, Aquitalea, Methylocella, Methylobacterium-Methylorubrum, Staphylococcus, Pseudomonas*, and *Acinetobacter*. Seven out of ten shared microbes constituted members of the ten most abundant ASVs found on the respective transporting frog (*Supplementary file 1h*).

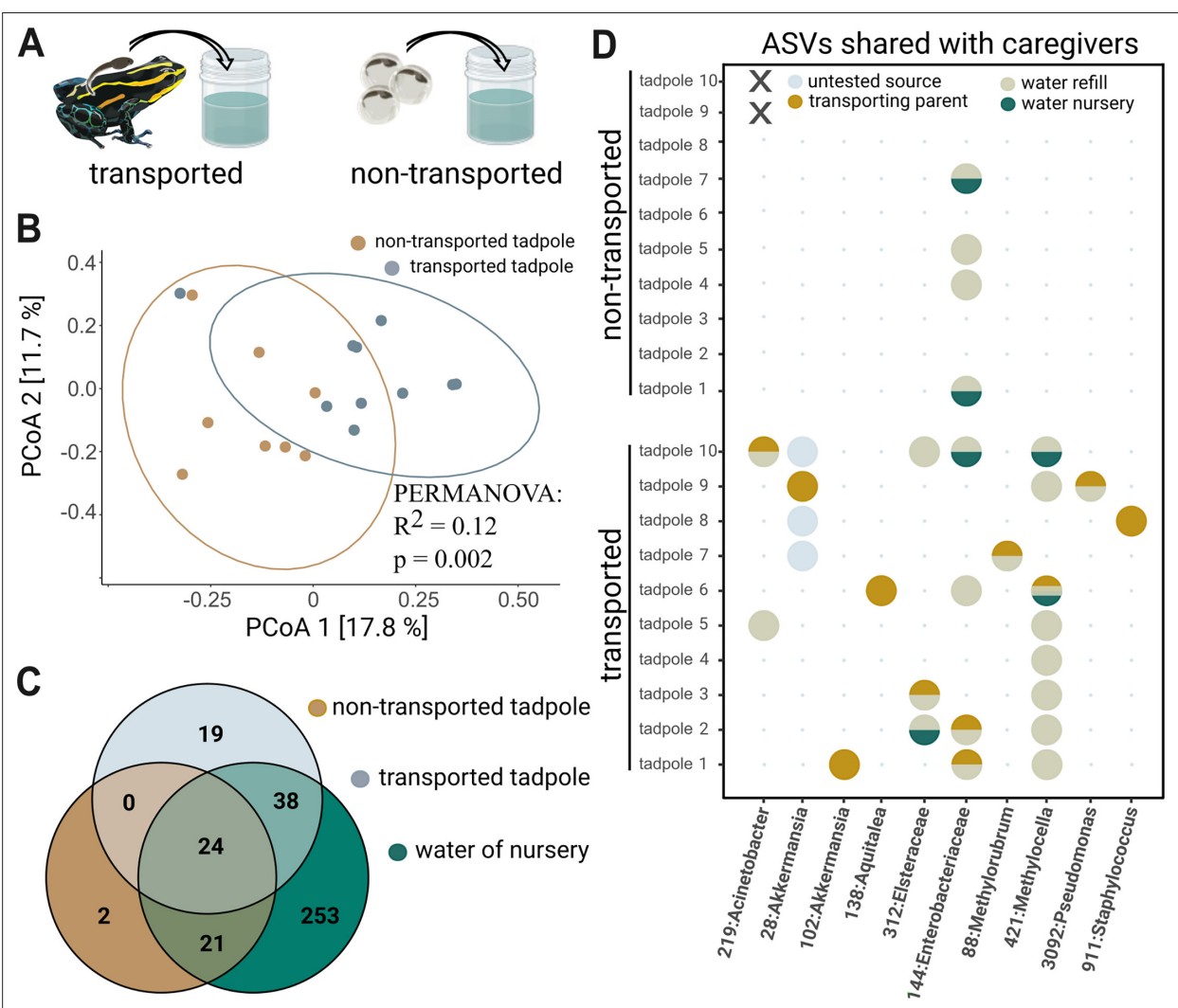

**Figure 4.** Tadpole transport influences community structure. (**A**) Tadpoles collected from the back of their caregiver ('transported,' N=10) and reared in artificial cups for one month were compared to 6-week-old tadpoles that hatched from eggs transferred to artificial cups and did not experience transport by adult frogs ('non-transported,' N=10). (**B**) Principal Coordinate Analysis constructed using unweighted Unifrac distances shows that transported tadpoles cluster significantly differently from non-transported tadpoles. Significances were determined with a PERMANOVA followed by a pairwise Adonis post hoc test. (**C**) Venn diagram comparing unrarefied Amplicon Sequence Variants (ASVs) agglomerated on a genus level between transported tadpoles, non-transported tadpoles, and the aquatic environment. (**D**) Bubble diagram depicting the presence (circle) or absence (dot) of 10 ASVs shared between parents and transported tadpoles as well as their possible source ('transporting parent,' 'nursery water,' 'refill water,' or 'untested'). Non-transported tadpoles (N=2) that died prior to sampling are indicated by 'X'.

Absolute abundances of shared ASVs likely originating from the parental source pool (as identified by Sourcetracker) after one month of growth ranged from 7804–172,326 sequence copies per tadpole (*Supplementary file 1i*). Finally, we determined the success of shared ASVs in colonizing transported and non-transported experimental tadpoles. Even though at least six (min: 6, avg: 6.38, max: 7) of the ten ASVs were consistently present in the aquatic environment of non-transported experimental tadpoles, only one (family *Enterobacteriaceae*) successfully colonized their skin (*Figure 4D*, not transported tadpoles; *Supplementary file 1j*).

## Discussion

In this study, we aimed to understand the main factors influencing the establishment of microbial communities associated with anuran tadpoles receiving parental care. To better understand sources contributing pioneering taxa, we first aimed to determine the ontogenetic window in which anuran embryos are colonized. Next, we tested if a specialized anuran parenting strategy, tadpole transport, transmits early colonizers. Then, we sampled natural populations of frogs to investigate if parental signatures persist into later ontogenetic stages in an ecologically relevant context. We found that substantial microbial colonization of the poison frog *Rv* occurs after embryos hatch from the vitelline envelope. Furthermore, we demonstrated through cross-fostering experiments that transporting frogs serve as a source of skin microbes for tadpoles. Next, we sampled sympatric species in a Neotropical habitat and revealed that tadpoles of poison frogs showed more diverse microbial communities than tadpoles of leptodactylid frogs. Finally, a field experiment in the wild population indicated that microbial communities of *Rv* tadpoles after four weeks of growth were not more similar to transporting caregivers than to unrelated adults, although some tadpoles did maintain isolated ASV that were shared with the transporting adults. Together, our results demonstrate that microbial colonization of a poison frog tadpole begins at an ontogenetic stage immediately preceding parental contact, that parental care in poison frogs facilitates the transfer of microbes to newly hatched tadpoles, and that signatures of these parentally transmitted microbes can persist, albeit at low prevalences, through the first month of tadpole development. This study shows that tadpole transport may serve as a mechanism to transmit host-adapted microbes, thereby filling a knowledge gap in our understanding of the function of parental care in anurans.

Early microbial colonizers have been shown to shape community structure of newborns in insects (*Jones et al., 2022*), mammals (*Blyton et al., 2022*; *Kaur et al., 2022*; *Sprockett et al., 2018*), fish (*Almany, 2004*), birds (*Chen et al., 2020*), and anurans (*Jones et al., 2024*; *Jones et al., 2023*) via priority effects (see *Debray et al., 2022*; *Fukami, 2015* for reviews). For example, inoculation of embryos has been shown to increase the relative abundance of beneficial taxa such as *Janthinobacterium lividum* in tadpoles (*Jones et al., 2024*), whereas efforts to introduce the same organism into established adult communities have not led to long-term persistence (*Bletz, 2013*; *Woodhams et al., 2016*). Thus, determining critical developmental periods for microbial colonization of host tissue is crucial to assess which sources contribute these pioneering taxa. We detected microbes in embryos of all stages but found the abundance to be considerably lower than in the surrounding jelly. Additionally, we found that substantial colonization of *Rv* tadpoles happens after hatching from the vitelline membrane, at an ontogenetic stage immediately preceding skin-to-skin contact with the caregiver. These results underscore that the opportunistic colonization of the anuran embryo is much more restricted than that of the surrounding jelly (*Hayes et al., 2009*; *Kueneman et al., 2014*; *Walke et al., 2011*), which harbors stable microbial assemblages (*Hughey et al., 2017*; *Walke et al., 2011*). Such restrictions might result from host-led selection as observed in marine mammals (*Switzer et al., 2023*) and/or a barrier function of the vitelline membrane that limits microbial access to the developing embryo. For example, the latter was described for the avian vitelline membrane, where it is mediated by antimicrobial proteins (*Guyot et al., 2016*; *Mann, 2008*; *Mine and Kovacs-Nolan, 2006*). The substantial microbial colonization that occurred after hatching suggests that the anuran vitelline envelope plays an important role in restricting microbial access to poison frog embryos until they are sufficiently developed to be transported by a caregiver. Those bacteria that are present in unhatched tadpoles might have been acquired prior to hatching, such as in the case of the amphibian symbiont *Oophila amblystomatis* (*Kerney et al., 2011*), which is acquired by oviductal transmission and grows on the inside of individual salamander and frog vitelline membranes. Alternatively, the vitelline membrane may function as a selective barrier, permitting the passage of certain taxa, with

hosts acting as additional filter to select suitable colonizers. Future studies should, therefore, address the presence of antimicrobial proteins, biofilms, and immune components in the anuran embryo and vitelline membrane to shed light on the mechanisms that regulate microbial colonization of anuran embryos.

Across species, newborns might acquire bacteria not only through transfer from environmental source pools and other hosts (e.g. arthropods: *Mushegian et al., 2018*, amphibians: *Rebollar et al., 2016*, mammals: *Bik et al., 2016*) but also through vertical transmission during parent-offspring interactions (reptiles: *Bunker et al., 2021*; fish: *Sylvain and Derome, 2017*; humans: *Wang et al., 2020*). Modes of parental inheritance involve indirect transmission, like contact smearing onto the egg surface during or after oviposition, as found in European firebugs (*Salem et al., 2015*), and direct social acquisition, as observed in social bees (*Kwong et al., 2017*; *Leftwich et al., 2020*). Previous studies in anurans have investigated vertical transmission of microbes during parental care in the context of indirect transmission during egg attendance without considering less widespread parenting strategies involving direct parental contact. These studies did not find evidence for parental inoculation, as hatched tadpoles supported bacterial communities very similar to adult frogs but independent of caregiver identity (*Hughey et al., 2017*). In contrast, we investigated vertical transmission in an anuran system where parenting involves direct skin-to-skin contact between tadpoles and their caregiver. We found that both transported and non-transported tadpoles in our cross-foster experiment shared microbial ASVs with each other, indicative of acquisition of microbes from the jelly upon hatching as demonstrated for other anuran clades (*Hughey et al., 2017*; *Jones et al., 2024*; *Warne et al., 2017*). After 6 hr of contact, transported tadpoles but not their siblings shared more microbes with the surrogate heterospecific frog than with the parental species, suggesting that microbes can transmit vertically during parental care in poison frogs. Thus, in this clade, both clutch environment and transporting caregivers can serve as a source pool for hatched tadpoles.

Most poison frogs transport tadpoles on their backs, but the mechanism of adherence remains unclear. Similar to natural conditions, tadpoles that are experimentally placed onto a caregiver's back also gradually adhere to the dorsal skin, where they remain firmly attached for several hours as the adult navigates dense terrain. Although transport durations were standardized, species-specific factors- such as microbial density at the contact site, microbial taxa identity, and skin physiology, such as moisture -might influence microbial transmission between the transporting frog and the tadpole. While these differences may have contributed to varying transmission efficacies observed between the two frog species in our experiment, none of these factors should compromise the correct microbial source assignment. We thus conclude that transporting frogs serves as a source of microbiota for transported tadpoles. However, further studies on species-specific physiological traits and adherence mechanisms are needed to clarify what modulates the efficacy of microbial transmission during transport, both under experimental and natural conditions. For example, evidence for duration-dependent microbial transmission is seen in humans, where recent research highlights a cumulative contribution of the paternal microbiome to the assembly of infants' microbial communities (*Dubois et al., 2024*). In anurans, the time that hatched tadpoles spend in contact with jelly-associated microbes or in skin-to-skin contact with parents might, therefore, affect the success and stochasticity of colonizers that are transmitted during parental care. These transport logistics vary between and within anuran species (*Altig and McDiarmid, 2007*; *Hanken, 1999*) and are closely linked to trade-off decisions of the caregivers, which often need to balance territory defense, mating, and caring for offspring (*Pašukonis et al., 2019*; *Ringler et al., 2013*; *Wells, 2007*). For instance, *Rv* males often assist tadpoles with hatching by tearing the capsule apart with their rear legs to initiate transport (*Brown et al., 2008*), but have also been observed to pick-up free-swimming tadpoles (*Schulte and Mayer, 2017*). Likewise, *Af* tadpoles have been observed to remain in the jelly for several days before being transported (*Peignier et al., 2022*; *Ringler et al., 2013*). Parentally transmitted microbes might thus constitute pioneering taxa in an unoccupied niche if caregivers assist with hatching or encounter established assemblages as in the latter scenario. Prolonged parental care has been shown to enhance the generational transmission of symbiotic microorganisms in burying beetles (e.g. *Körner et al., 2023*) and thus, duration of the skin-to-skin contact during tadpoles transport might influence microbial acquisition in a time-dependent manner. How duration of tadpole transport, which ranges from a few hours to several days depending on the species (*Lötters et al., 2007*; *Hanken, 1999*; *Pašukonis et al., 2019*), affects stochasticity of transmitted microbes remains unexplored. Collectively, our findings

suggest that early life microbes are introduced to the poison frog host in a mixed transmission mode, where community members can be sourced environmentally from the egg jelly and socially from a caregiving frog. Recent evidence from Kouete et al. documented the first case of vertical transmission in amphibians, observed in subterranean caecilians that protect and feed their offspring *Kouete et al., 2023*. To our knowledge, ours is the first study that extends evidence for vertical transmission during parental care to the order of anurans.

To investigate if parental signatures persist into later ontogenetic stages, we tested the hypothesis that a history of direct contact would result in more similar skin communities between tadpoles and caregivers relative to unrelated adults. Based on previous findings which documented that priority inoculation of tadpoles increased the relative abundance of some beneficial inoculates (*Jones et al., 2024*), we hypothesized that vertical transmission manifests in persistent associations with parentally acquired microbes. Similar to Hughey et al., we found that vertical transmission did not lead to a higher degree of similarity between caregiver and tadpole skin microbiota relative to unrelated adult-tadpole pairs. Likewise, communities associated with adults and tadpoles of transporting species were no more similar than those of non-transporting species. While poison frog tadpoles do acquire caregiver-specific microbes during transport, most of these microbes do not persist on the tadpoles' skin long-term. This pattern can likely be attributed to the capacity of tadpole skin and gut microbiota to flexibly adapt to environmental changes (*Emerson and Woodley, 2024*; *Santos et al., 2023*; *Scarberry et al., 2024*). It may also reflect the limited compatibility of skin microbiota from terrestrial adults with aquatic habitats or tadpole skin, which differs structurally from that of adults (*Faszewski et al., 2008*). As a result, many transmitted microbes are probably outcompeted by microbial taxa continuously supplied by the aquatic environment. Interestingly, microbial communities of the non-transporting *Ll* were more similar to their adult counterparts than those of poison frogs. This pattern might reflect differences in life history among the species. Adult *Ll* commonly inhabit the rock pools where their tadpoles develop, but adults of the two poison frog species visit tadpole nurseries only sporadically for deposition. These differences in habitat use may result in adult *Ll* hosting skin microbiota that are better adapted to aquatic environments as compared to *Rv* and *Af*. Additionally, their presence in the tadpoles' habitat could make *Ll* a more consistent source of microbiota for developing tadpoles. Transient shifts in the composition of tadpole-associated communities followed by later convergence have also been observed in wood frogs experimentally inoculated with bullfrog gut microbes (*Warne et al., 2019*). Notably, despite this compositional convergence, effects of early microbial disruption on host physiology and disease susceptibility persisted- an outcome also documented in fruit flies (*Cox et al., 2014*). Therefore, it is possible that vertically acquired microbes shape communities of poison frogs in different ways than by gaining abundance priority on tadpole skin. For example, they may durably imprint the immune system (*Fallet et al., 2022*; *Gensollen et al., 2016*) and alter host metabolic phenotypes (*Cox et al., 2014*; *Sommer and Bäckhed, 2013*). Additionally, isolated ASVs contributing to central genera of communities on adults persisted on some tadpoles in varying abundance. There is clear evidence that low-abundant keystone taxa, despite their rarity, can act as drivers of compositional changes (*Han and Vaishnava, 2023*). For instance, low abundance ASVs drive compositional changes in the hindgut of subterranean termites after dietary alterations (*Benjamino et al., 2018*) and confer resistance to Salmonella-induced colitis in mice (*Herp et al., 2019*). Moreover, despite their abundance in the water of all tadpole nurseries, most of the ASVs that tadpoles shared with caregivers failed to colonize non-transported hosts. These results suggest that vertical transmission may be a more effective mode of transmission of host-adapted microbes compared to horizontal transmission from the environment. Whether vertically transmitted microbes shape the anuran immune response and if retaining certain ASVs allows them to become dominant community members after metamorphosis remains to be investigated.

We found that the higher diversity of communities associated with tadpoles of species that exhibit offspring transport was not linked to the diversity found in tadpoles' aquatic habitat but might instead be associated with other life history traits. Although skin alkaloids are found on many adult poison frogs including *Rv* and influence skin community composition (*Caty et al., 2024b*), they are not present on adult *Af* or on tadpoles of any species included in this study (*Villanueva et al., 2022*) and are thus unlikely to cause the observed differences in diversity. It is possible that oviposition strategy (*Af* and *Rv* lay terrestrial egg clutches, whereas *Ll* lays eggs directly in water) or physiological characteristics like skin pH or abundance of antimicrobial proteins (*Conlon, 2011*; *Faszewski et al., 2008*;

*Kueneman et al., 2014*) contributed to increased diversity in *Rv* and *Af* relative to *Ll* tadpoles. Alternatively, transport might have been co-opted in an evolutionary context to increase microbial diversity in tadpoles and shape immune recognition and health in the anuran host in a manner comparable to humans (*Al Nabhani and Eberl, 2020*; *Donald and Finlay, 2023*; *Gensollen et al., 2016*). While increased microbial diversity is not inherently advantageous, it has been associated with beneficial outcomes such as improved immune function, lower disease risk, and enhanced fitness in multiple other vertebrate systems. For example, higher diversity of gut microbial communities is associated with enhanced exploratory behavior in songbirds (*Florkowski and Yorzinski, 2023*) and improved pathogen resilience in fish (*de Bruijn et al., 2018*). Previous research in frogs has linked higher diversity of tadpole skin communities to enhanced parasite resistance later in life (*Knutie et al., 2017*; *Warne et al., 2019*). However, increased microbial diversity is not a known outcome of vertical transmission, and further studies across a broader range of transporting and non-transporting species are needed to assess the role of transport in shaping diversity of tadpole-associated microbial communities. Moreover, the low chytrid infection-rate in the studied natural populations limits the ability to draw conclusions about differences in disease susceptibility within and between species or life stages in our dataset.

Elucidating the key factors that select for parental care and account for the diversity of care strategies across various species remains a challenge in the field of evolutionary and behavioral ecology (*Balshine, 2012*; *Clutton-Brock and Scott, 1991*; *Gross, 2005*; *Balshine, 2012*). Microbiota management is documented to actively drive, rather than merely accompany sociality and parenting behavior in other organisms like carrion feeding insects (*Biedermann and Rohlfs, 2017*; *Körner et al., 2023*) and has recently been proposed as a factor driving the evolution of paternal care in vertebrates (*Gurevich et al., 2020*). Even though commensal skin microbiota undoubtedly play a pivotal role in the health and pathogen defense of amphibians, it remains poorly understood how complex parental care strategies in this class aid microbial transmission. This is especially surprising as amphibians have recently been identified as the most threatened class of vertebrates (*Daszak et al., 2003*; *Luedtke et al., 2023*; *Wake and Vredenburg, 2008*) due to their susceptibility to globalization-related threats like climate change and disease-spread. Our observation that anuran parental care transmits microbes fills a knowledge gap about the function of parental care in this class and highlights the need for further studies across more anuran species with contrasting reproductive strategies.

## Methods

### Captive-bred animals

Captive-bred *Rv* of our laboratory colony are kept non-sterile and reproduce year-long. Males attend the terrestrial clutch of 3–5 eggs and transport larva to water when they reach Gosner stage 24–25 where they hatch from the vitelline membrane. *Rv* were housed in pairs and provided with horizontally mounted film canisters as egg deposition sites, and film canisters filled with water (treated with reverse osmosis R/O Rx, Josh's Frogs, Owosso, MI) for tadpole deposition (*Goolsby et al., 2023*).

### Microbial colonization of embryos and egg jelly

We used individually packed sterile transfer pipettes (Samco, Thermo Scientific, 7.7 mL, cat # 202-1SPK) with a cut tip to move eggs and embryos. All tools were treated with 8.25% Sodium Hypochlorite (CloroxPro) and wiped down with 70% Ethanol before use and between tissues and tadpoles. The experimenter always wore gloves that were changed between individuals and disinfected with 70% Ethanol before handling frogs.

Triplicates of five different developmental stages (*Gosner, 1960*) before (stages 10, 18, 23, 24) and after hatching from the vitelline membrane (stages 24, 25) were collected from three different breeding pairs. We manually separated the jelly from the egg sac with autoclaved forceps, a procedure commonly done in preparation for microinjections. The embryo in the vitelline membrane was transferred to a small petri dish with sterile, filtered deionized water and opened using forceps. Tadpoles were transferred to a new petri dish with clean water twice to remove transient bacteria from the skin and were then euthanized and homogenized. Organic matter was removed from the jelly, and both jelly and tadpole tissues were flash frozen and stored at –80°C until DNA isolation. Isolation of DNA was performed as detailed below, and 1 μL of isolated DNA was used for PCR amplification of

the full length 16 S rRNA region (~1500 bp) with 1 µL of 10 µM forward 27 F (5'-AGAGTTTGATCM TGGCTCAG-3') and reverse primer 1492 R (5'-TACGGYTACCTTGTTAYGACTT-3') (*Fredriksson et al., 2013*), 12.5 µL of OneTaq Hot Start Quick Load polymerase (New England Biolabs) and 9 µL water. We followed an amplification protocol with denaturation for 30 s at 94°C followed by 30 cycles of denaturation for 15 s at 94°C, annealing for 30 s at 58°C and extension for 2 min at 68°C, and a final extension period of 5 min at 68°C. To determine the presence or absence of amplification, we visualized 5 µL of each PCR product on a 1% Agarose gel (90 mV for 90 min) with 2 µL of 1 kb GeneRuler for sizing.

## Quantitative analysis of 16S rRNA copy numbers with droplet digital PCR (ddPCR)

Quantification of prokaryotic concentration for each sample was determined by ddPCR for all samples (*Doyle et al., 2025*). Sample DNA was serially diluted 1:10 in sterile nuclease-free water to dilute the concentration of any PCR inhibitors using a liquid handler (Agilent, Agilent Velocity 11 Vprep). 16 S rRNA concentrations from embryonic tissue are reported from undiluted DNA as its concentration was approaching the limit of detection in diluted reactions. Universal 16 S rRNA primers (331 F/797 R) and 16 S rRNA HPLC-purified FAM probes were used, as previously described (*Langenfeld et al., 2021*). Primer-probe mixture was created by mixing forward primer, reverse primer, and probe 1:1:1 for a total volume of 0.264 µL per reaction. Each ddPCR reaction was composed of 11 µL of ddPCR Supermix for probes (Bio-Rad, 1863024), 0.264 µL of primer-probe mixture, 4.736 µL of sterile nuclease-free water and 6 µL of sample, for a total of 22 µL/reaction. As a positive control, equal amounts of NIST Microbial Pathogen DNA Standards for Detection and Identification (NIST, RM8376) components A through R were combined in equal amounts. Each ddPCR plate included a positive control, NIST mixture, and negative control, sterile nuclease-free water, in quadruplicate. ddPCR reactions were performed with the QX200 AutoDG Droplet Digital PCR system (Bio-Rad). PCR amplification was performed with the Bio-Rad T100 thermocycler using the following program: 95°C for 10 min, 40 cycles of 95°C for 30 s and 56°C for 1 min and 72°C for 2 min, followed by one cycle of 4°C for 5 min and 95°C for 5 min with a ramp speed of 2°C/s at each step. Amplified reactions were quantified using a ddPCR reader. Thresholds were set for each ddPCR reaction based on the negative and positive control rain plots generated by the QX Manager Software, 2.1 Standard Edition (Bio-Rad). All reactions had greater than 15,000 accepted droplets. 16 S rRNA copies per droplet were calculated by taking the natural logarithm of the ratio of accepted droplets to negative droplets for each reaction. Values were corrected for droplet volume and ddPCR reaction volume to calculate total 16 S rRNA copies per reaction. We further calculate and report the total 16 S rRNA copies per µL by accounting for the sample ddPCR volume, dilution factor, and DNA extraction volume. Differences in copy number concentrations detected in jellies, hatched and unhatched tadpoles were determined using a Kruskal Wallis test with p-values adjusted for multiple comparisons using Benjamini Hochberg correction (*R Core Team, 2023*) and followed by a Dunn test.

## Vertical transmission of bacteria

Cross-fostering tadpoles onto non-parental frogs has been used previously to study navigation in poison frogs (*Pašukonis et al., 2017*). According to our experience, successful adherence to both parent and heterospecific frogs depends on the developmental readiness of tadpoles, which must have retracted their gills and be capable of hatching from the vitelline envelope through vigorous movement. Another factor influencing cross-fostering success is the docility of the frog during initial attachment, as erratic movements easily dislodge tadpoles before adherence is established. *Rv* are small, jumpy frogs that are easily stressed by handling, making experimental fostering of tadpoles-even their own- impractical. Therefore, we favored an experimental design where tadpoles initiate natural transport, and parental frogs pick them up with a 100% success rate. We chose the poison frog *Os* as foster frog because adults are docile, parental care in this species involves transporting tadpoles, and skin microbial communities differ from *Rv*- a critical prerequisite for our Sourcetracker analysis. The use of the docile *Os* as the foster species enabled a 100% cross-fostering success rate, with no notable differences in adherence strength of tadpoles after 6 hr.

Siblings of one clutch were randomly assigned to one of three treatments: (1) no transport, (2) transport by the biological caregiver, or (3) transport by the heterospecific surrogate frog *Os*. Tanks of *Rv* breeding pairs were monitored for clutches of at least three eggs daily. Embryos assigned to group

one or three were placed into a petri dish in a separate tank without access to adults when tadpoles reached a developmental stage close to hatching (Gosner stage 22) to bypass transport by the biological father. The remaining eggs in the parental tank were monitored with security cameras (Wyze v3, *Goolsby et al., 2023*) that notify the user of any movement in the canister. After tadpole pickup by the biological father, all water canisters for tadpole deposition were removed from the tank to standardize transport time to 6 hr for both groups. Siblings of group three were transferred to a new petri dish, hatched using a sterile brush, and directly transferred to the back of a surrogate frog (as described in *Pašukonis et al., 2017*) that was caught in a fresh plastic bag and rinsed with sterile water. The surrogate frog was placed into a plastic containment (Sterilite 16428012) with moist, autoclaved paper towels for the duration of the experiment. After 6 hr, the transporting biological father and the surrogate frog were caught in a fresh plastic bag. Each tadpole was removed from the back and washed twice to remove transient bacteria by transferring them to new petri dishes with 10 ml sterile water. Non-transported tadpoles were removed from the jelly and washed in a similar way. Each frog was rinsed with sterile water, and a skin swab of their back was collected using a sterile Puritan applicator (Puritan Medical Products, 25–206 1PD BT) (*Caty et al., 2024b*). All tadpoles were euthanized, their skin was collected, and skin and applicators were stored at –20°C until DNA extraction. We chose to sample whole skin instead of swabs to detect small amounts of transferred bacteria on and in the skin.

## Study species, reproductive strategies, and life history

*Oophaga sylvatica (Os)* (Funkhouser, 1956; CITES Appendix II, IUCN Conservation status: Near Threatened) is a large, diurnal poison frog (Family *Dendrobatidae*) inhabiting lowland and submontane rainforests in Colombia and Ecuador. While male *Os* care for the clutch of up to seven eggs, females of this species transport 1–2 tadpoles at a time to water-filled leaf axils where tadpoles complete their development (*Pašukonis et al., 2022*; *Silverstone, 1973*; *Summers, 1992*). Notably, females return regularly to these deposition sites to provision their offspring with unfertilized eggs.

*Ranitomeya variabilis (Rv)* (Zimmermann & Zimmermann 1988; CITES Appendix II, IUCN Conservation status: Least Concern) is a small diurnal poison frog (Family *Dendrobatidae*) with male uniparental care that lives largely arboreal and uses bromeliads as a resource for reproduction. Adult frogs defend territories, are polygamous and typically lay clutches of 3–4 eggs into small arboreal bromeliads (*Catopsis berteroniana*) that are abundant on Clusia trees in the rock savanna. Male frogs transport hatched tadpoles to individual water bodies that form in the leaf axils of the large terrestrial tank bromeliads (*Aechmea aquilega*). The cannibalistic tadpoles grow up in individual pools of ~80 ml (*Poelman et al., 2013*) where they feed on algae and leaf debris until they complete their development after about 3–5 months (*Poelman and Dicke, 2007*).

*Allobates femoralis (Af)* (Boulenger, 1884; CITES Appendix II, IUCN Conservation status: Least Concern) are polygamous diurnal poison frogs (Family *Aromobatidae*) that inhabit the understory of primary tropical forests. Females are attracted by advertisement calls of the territorial males and deposit clutches of 11–25 eggs in the leaf litter (*Fischer et al., 2020*; *Stückler et al., 2019*; *Weygoldt, 1980*). Fathers care for their offspring by attending clutches and shuttle hatched-tadpoles to water bodies where the social tadpoles grow up sharing a pool (*Lescure, 1976*; *Ringler et al., 2013*; *Ringler et al., 2018*).

*Leptodactylus longirostris (Ll)* (Boulenger, 1882; IUCN Conservation status: Least Concern) are medium sized nocturnal leptodactylid frogs (Family *Leptodactylidae*) that inhabit the rock outcrops. They shelter in rock crevices and ground covering vegetation during the day and dwell in seasonal rock pools on the Inselberg plateau during nights. They breed in the rainy season and lay their eggs in ephemeral rock pools. After clutch deposition, they do not care further for their tadpoles, which collectively grow up in the deposition pond and feed on algae and plant debris.

## Field site and natural populations

The natural study site of *Rv* and *Ll* is situated on top of the mountain 'Inselberg' in vicinity to the Centre Nationale de la Recherche Scientifique managed research station (4°5' N, 52°41' W) within the Nature Reserve 'Les Nouragues' in French Guiana (*Bongers et al., 2011*). Patches of Clusia trees are separated by bare granite rocks and exposed to extreme environmental conditions (e.g. temperature oscillations between 18°C–75°C; *Sarthou, 2001*). *Rv* and *Ll* (IUCN Conservation status: Least Concern)

inhabit the granite outcrop of the study site, *Af* inhabit forest floors of primary terra-firme forests surrounding the campsites 'Inselberg' and 'Saut Pararé' (4°02'N/52°41'W) in the Nouragues reserve.

## Sampling of frogs in the field

Frogs were caught in fresh plastic bags upon encounter, rinsed with sterile water to remove transient bacteria and soil particles and then swabbed with 20 strokes each on the dorsal and lateral sides as well as on each leg and between the toes using Sterile Polyester Applicators (Puritan Medical Products, 25–206 1PD BT). Due to limitations in the availability of dry ice at the remote study site, applicators were directly transferred into 800 µL of buffer CD1 (a lysis buffer containing chaotropic salts) of the Qiagen PowerSoil Pro extraction kit and frozen until DNA extraction. Previous studies have found no difference in OTU richness and evenness between native frozen and lysis buffer-stored swabs (*Hallmaier-Wacker et al., 2018*).

## Field sampling of tadpoles and their environment

To avoid disturbing microbial communities of tadpoles and their habitats before sampling, free-swimming tadpoles were visually categorized as 'small' (with a body length of less than 6 mm, typically Gosner stages 25 and 26), 'medium' (with a body length exceeding 6 mm but no visible dorsal pattern, stages 26–33), 'large' (visible yellow dorsal pattern, stages 34–41) or 'metamorph' (front limbs visible, stages 42–46). Tadpoles of stages 29–41 (Gosner) were collected from bromeliad pools for swabbing when they at least doubled the body size of freshly hatched tadpoles, at the earliest four weeks after deposition. The water and sediment of the nursery was sampled prior to tadpole extraction by submerging a sterile applicator tip into the leaf axil pool and moving it over all plant material contained in the water for 10 s. *Rv* tadpoles were extracted from bromeliad leaf axils with a custom-designed vacuum extractor consisting of an inlet hose wide enough to allow the passage of a late-stage tadpole and a suction hose with air filter, both tightly connected to the lid of a one-liter bottle. The inlet hose was submerged into the bromeliad pool while applying suction to the second hose to create a vacuum that transferred the pool content into the collector. After each use, all parts of the collector were cleaned with soapy water and rinsed, followed by a sodium hypochlorite disinfection for 1 hr and a rinse in 70% Ethanol as described for the dissection tools. After collection, tadpoles were transferred to a petri dish with sterile water twice, using individual sterile transfer pipettes with a cut tip. For swabbing, tadpoles were collected in the bulb of a new sterile pipette. The water was discarded, and the tadpole was swabbed on all body parts with a sterile applicator tip by moving the tip over its body for 10 s. We further processed swabs in the same way as adult swabs. *Af* tadpoles of stage 25–26 (Gosner) were collected from an artificial pool in the vicinity of camp 'Saut Parare.' *Ll* tadpoles of Goser stage 34–41 were collected from three natural pools in the study plot 'Inselberg.' We sampled triplicates of each pool containing multiple tadpoles. All tadpoles were processed for swabbing in a similar way as *Rv* tadpoles.

## Field experiment to compare transported and non-transported *Rv* tadpoles

To directly compare parental frogs with transported and non-transported tadpoles while reducing predation and variation in the quality of their nurseries, we reared 10 tadpoles that were collected from the back of eight caregivers and ten tadpoles that hatched from eggs without experiencing transport in polypropylene cups (USP #77876, 60 ml). Each cup was punctured at the 50 ml water level, covered with a sterilized mosquito net secured with a rubber band to minimize predation, and attached to the stem of a Clusia tree. All tadpoles received water collected from bromeliads as their initial aquatic environment with Clusia leaves as shelter and food. Cups were exposed to rain and during dry periods, rainwater was collected and mixed with leaf litter to refill cups. Triplicates of water samples were obtained using the procedure described for bromeliad water. We caught transporting frogs upon encounter, transferred tadpoles from their backs to cups, and swabbed the parenting frog as described above. To avoid transport, eggs from five clutches were transferred to cups before tadpoles hatched. Tadpoles were sampled after growing for at least 26 days (min: 26, max: 77), when all tadpoles of a group exceeded ~5 mm in body size, as smaller individuals exhibited increased mortality after swabbing in previous lab experiments. After swabbing, tadpoles were measured and photo documented.

## DNA extraction and 16S rRNA gene sequencing

The Qiagen PowerSoil Pro Kit was used to extract DNA from all swabs and tissues. The protocol was adapted for use with swabs as described previously (*Caty et al., 2024a*). DNA concentrations were quantified using a Qubit. Samples were pooled by volume and the V4 region of the 16 S rRNA gene was amplified using 515 F (GTGYCAGCMGCCGCGGTAA) and 806 R (GGACTACNVGGGTWTC TAAT) primers (*Bletz et al., 2017*) and barcoded using standard Illumina unique dual indices (UDIs). We performed two separate sequencing runs: Laboratory collected samples were sequenced in a 2x300 nt paired-end configuration on an Illumina MiSeq v3 run, field collected samples in a 2x250 nt paired-end configuration on a NovaSeq 6000 SP Flowcell (Roy J Carver Biotechnology Center, University of Illinois).

## Sequence processing and bioinformatics

We annotated in-line barcodes based on the first 7 bases of each sequencing read (umi-tools, *Smith et al., 2017*), split out reads that matched each known barcode combination (grep, *Project GNU, 1998*), trimmed the remaining primer sequences from the sequencing reads (cutadapt, *Martin, 2011*) counted the number of sequencing reads in each file and removed files (including negative controls) with low read numbers (<100 reads) from the dataset. We processed the remaining reads with the R Divisive Amplicon Denoising Algorithm package 'dada2' (version 1.28.0) (*Callahan et al., 2016*). Taxonomy was assigned using the Silva 138 database (Ref NR99) (*Quast et al., 2013*). The count table, taxonomy table, and sample-associated data were integrated into a phyloseq object using the R package 'phyloseq' (version 1.44.0) (*McMurdie and Holmes, 2013*) for further data analysis. Any ASVs detected in control sequencing reactions without DNA as well as ASVs with a phylum designation that was 'NA,' Eukaryotic, belonging to the family of Mitochondria or the class of Chloroplasts were removed from the datasets. Samples with very low reads (clearly below the first quartile) were excluded from each group.

## ITS sequence processing and bioinformatics

Amplification targeted the second internal transcribed spacer (ITS2) region of the fungal ribosomal cistron using primers ITS3 and ITS4 (*White et al., 1990*). Full primer sequences included Illumina Nextera transposase adapters (underlined) and were as follows: ITS3-5'-TCG TCG GCA GCG TCA GAT GTG TAT AA GAG ACA GGC ATC GAT GAA GAA CGC AGC-3' and ITS4-5'-GTC TCG TGG GCT CGG AGA TGT GTA TAA GAG ACA GTC CTC CGC TTA TTG ATA TGC-3'. Reaction mixes contained 1 x AccuStart II PCR SuperMix (Quantabio), 0.4 µM forward and reverse primers, and 2 µl of template DNA. The thermal cycling regime consisted of an initial denaturation and enzyme activation step at 94°C for 3 min, followed by 30 cycles of 94°C for 45 s, 50°C for 60 s, and 72°C for 90 s, with a final extension step at 72° for 10 min. Ten additional PCR cycles were used to tag diluted amplicons with unique dual indices (Illumina; cycling regime as above but with annealing at 54°C). Amplicons were pooled, cleaned using AMPure XP beads (Beckman Coulter), and sequenced on an Illumina MiSeq instrument (2×300 nt) at the DNA Services Lab, Roy J. Carver Biotechnology Center, University of Illinois at Urbana-Champaign. Reads processing was consistent with the DADA2 ITS workflow (https://benjjneb.github.io/dada2/ITS_workflow.html). Raw reads were trimmed of non-biological sequence using cutadapt (*Martin, 2011*). DADA2 was used to filter and truncate the reads, infer amplicon sequence variants (ASVs), and remove chimeras (*Callahan et al., 2016*). Taxonomy was assigned using DADA2's implementation of the RDP naïve Bayesian classifier (*Wang et al., 2007*) against a UNITE reference database (version 9.0; *Abarenkov et al., 2023*). The ASV count table, taxonomy table, and sample-associated data were integrated into a single object in R using the phyloseq package (*McMurdie and Holmes, 2013*).

## Microbiota composition analyses

All statistical analyses were performed in R (v 4.3.2) (*R Core Team, 2023*), and 'ggplot2' (v 3.4.4) (*Wickham, 2016*) was used for visualizations if not stated otherwise. Illustrations of all species were created in Adobe Illustrator (v 28.3) (*Adobe Inc, 2024*) and all figures were compiled in InkScape (v 1.1) (*Inkscape Project, 2021*).

### Microbiota composition analyses of captive frogs and source tracking

First, we calculated beta diversity between skin communities of adult frogs that served as transporting frogs to determine if communities are distinct enough to distinguish them as microbial source

communities. We applied Principal Coordinates Analysis (PCoA) using the package 'phyloseq' to visualize variation in the genus-agglomerated dataset based on two dissimilarity indices: Bray-Curtis, a widely used measure considering relative abundances, and unweighted Unifrac, a presence-absence measure of ASVs based on phylogenetic distances. Data were converted to relative abundances before calculating Bray-Curtis or rarefied using a rarefaction over 300 iterations function phyloseq_mult_raref_avg in the package 'metagmisc' (*Mikryukov, 2023*) before calculating Unifrac distances. We confirmed equal dispersion between compared groups using the betadisper function of the package 'vegan' (v2.6.4) *Oksanen et al., 2022* followed by a Tukey post-hoc test. Variation was analyzed with a PERMANOVA (function 'adonis' within the package 'vegan') using Bray-Curtis and unweighted Unifrac dissimilarities as responses and species or transporting species as predictors. Significant PERMANOVAs were followed by pairwise multilevel post-hoc comparisons between groups with the function pairwise.adonis (package 'pairwiseAdonis,' *Arbizu, 2017*). We applied the same method to compare if communities between tadpoles that had been transported by a homospecific frog (*Rv*) or a heterospecific frog (*Os*) differed after 6 hr of tadpole transport. We further used 'Sourcetracker' (*Knights et al., 2011*) to determine whether transporting frogs served as a source for bacteria detected on tadpole skin after 6 hr of transport. This Bayesian approach was developed to estimate proportion of contaminants in a given community that come from possible source environments and is commonly used to determine if parental communities serve as source for offspring communities (*Kouete et al., 2023*; *Murphy et al., 2023*; *Switzer et al., 2023*). We performed two sets of analysis to determine the source proportions of adult *Rv* and *Os* in tadpole communities. First, we defined transporting frogs of both species as possible microbial sources and all tadpoles, including non-transported controls as sinks. Then, we defined transporting frogs and non-transported control tadpoles as possible sources to visualize the proportion of bacteria acquired in the clutch prior to transport. For both approaches, we rarefied the source dataset of transporting frogs to decrease the influence of high-coverage source samples on the analysis. For the first approach, we plotted the source proportions of *Rv* and *Os* communities for non-transported tadpoles, tadpoles transported by *Rv*, and tadpoles transported by *Os*, and compared between the experimental conditions for each source (*Rv* or *Os*) separately using a Kruskal Wallis test, with p-values adjusted for multiple comparisons using Benjamini Hochberg correction (*R Core Team, 2023*). Additionally, differences in community proportions shared with the transporting species ('transporting frog') or the non-transporting species ('non-transporting frog') were determined for non-transported and transported tadpoles and compared with the same analysis. For the second approach, we determined the source proportions that transported tadpoles had acquired from the clutch, from *Rv* adults, and from *Os* adults, and we determined differences between tadpoles transported by *Rv* or by *Os* using a paired Wilcoxon test with Benjamini-Hochberg correction (*R Core Team, 2023*).

To assess training quality, we evaluated model self-assignment using source samples. We selected the model trained on a dataset rarefied to the read depth of the adult frog sample with the lowest read count (48162 reads), as it showed the best overall self-assignment performance, whereas models trained on datasets rarefied to the lowest overall read depth performed worse. Unlike studies using technical replicates, our source samples represent distinct biological individuals and sampling timepoints, where natural microbiome variability is expected within each source category. Consequently, we considered self-assignment rates above 70% acceptable. All source samples were correctly assigned to their respective categories (*Rv*, *Os*, or control), but with varying proportions of reads assigned as 'Unknown'. Adult frog sources were reliably self-identified with high confidence (*Os*: 97.2% median, IQR = 1.4; *Rv*: 76.3% median, IQR = 38.1). Adult *Rv* frogs displayed a higher proportion of 'Unknown' assignments compared to *Os*, reflecting greater biological variability among individuals and/or a higher proportion of rare taxa not well captured in the training set. The control tadpole source showed lower self-assignment accuracy (median = 30.5%, IQR = 17.1), as expected given the low microbial biomass of these samples, which resulted in low read depth. Low read-depth limits the information available to inform the iterative updating steps in Gibbs sampling and reduces confidence in source assignments. We, therefore, verified the robustness of our results by performing the second Sourcetracker analysis as described above, training the model only on adult sources and assigning all tadpoles, including low-biomass controls, as sinks (as described above). Self-assignment rates for the second training set varied (*Os*: 79.2% median, IQR = 29; *Rv*: 96.6% median, IQR = 3.7), while results remained consistent across analyses, supporting the reliability of our findings.

## Microbiota composition analyses of wild populations

We calculated mean and standard deviation across all samples per group for the number of phyla and families from a rarefied dataset to compare groups. The total number of unique phyla and families for each group was calculated from the unrarefied dataset to include low abundance taxa. We plotted the relative abundance of taxa agglomerated at the phylum level with an abundance greater than 2%. In addition, we identified the most abundant genera (N=10) for each group based on overall mean relative abundances. The number of *Batrachochytrium*-inhibiting taxa was calculated by blasting the rarefied dataset against the AmphiBac-Database-2023.2 (*Woodhams et al., 2015*; Antifungal database: AmphibBac_InhibitoryStrict_2023.2; https://github.com/AmphiBac/AmphiBac-Database; *Bletz, 2025*) and calculating the average across all samples of a group.

All alpha diversity analyses were conducted with datasets rarefied to 90% of the read number of the sample with the fewest reads in each comparison and visualized with boxplots. We compared richness, Shannon index, and evenness between different life stages of the same species, as well as between tadpoles from different species and their environments. Differences in alpha diversity measures between groups were determined using an ANOVA (for normally distributed, homoscedastic data) followed by a Tukey post-hoc test or Kruskal-Wallis (for non-normally distributed data) followed by a Dunn test. P-values were adjusted for multiple comparisons using Benjamini-Hochberg corrections.

We determined beta diversity across frog species, life stages, and aquatic environments using two commonly used dissimilarity measurements as described for the captive-bred dataset. ASVs found in only one frog were removed from the full, non-rarefied dataset. Variation was analyzed with a PERMANOVA (function adonis within the package 'vegan'), using Bray-Curtis and unweighted Unifrac dissimilarities as responses and species, life stage, species-life stage interaction, and parenting behavior as predictors. Significant PERMANOVAs were followed by pairwise multilevel post-hoc comparisons between groups with the function pairwise.adonis (package 'pairwiseAdonis,' *Arbizu, 2020*).

As previously suggested (*Neu et al., 2021*), we determined genus level agglomerated taxa between adults, tadpoles and aquatic environments from unrarefied data converted to relative abundances across different prevalence (prev) and abundance (abd) cutoffs. This approach avoids using arbitrary thresholds to define a taxonomic 'core.' The following cutoffs were evaluated using the function 'core_members' of the package 'microbiome' (v1.22.0, *Lahti and Shetty, 2012*; *Lahti and Shetty, 2023*): (1) prev 100% without abd cutoff, (2) prev >75% with abd >1%, (3) prev >75% with abd >0.1% and (4) prev >75% without abd cutoff. We listed the most abundant genera for each species (*Rv*, *Af,* and *Ll*) and life stage (adult or tadpoles) with respect to their presence or absence in the core of the aquatic environment of the respective species and used the package 'eulerr' (*Larson, 2022*) to visualize overlaps of genera with prevalence over 75% and relative abundance over 0.1% with a Venn diagram.

To test whether community distances between tadpoles and adults are smaller in poison frogs than in a species without parental care, we performed a beta regression with an underlying logit function (R package 'betareg,' *Zeileis et al., 2021*) to analyze unweighted Unifrac distances of a rarefied dataset. This approach is suited to model predictions on our bounded distance dataset with high variance. ASVs were agglomerated at the genus level prior to calculating distances between tadpoles and adults of different species, while distances between related and unrelated pairs were calculated on a ASV level. To account for differences in samples size between groups, we performed iterated analyses with random subsamples of equal sample size (N=8 for distances between adults and tadpoles between species and N=10 for related vs. not-related pairs of *Rv*) and report averaged p-values (following the approach of *Hughey et al., 2017*).

## Microbiota composition analyses of wild experimental tadpoles

We determined and illustrated alpha and beta diversities of microbial communities on transported and non-transported tadpoles and significant differences between the two groups as described for wild populations. Variation was analyzed with a PERMANOVA (function adonis within the package 'vegan'), using unweighted Unifrac dissimilarity measures calculated from a rarefied dataset. Due to the limited sample size and low biomass of some tadpoles, we chose to work with relative ASV abundances rather than rarefaction to determine disparities and overlaps between the groups and their aquatic environment, following previous examples (*McMurdie and Holmes, 2013*; *Prest et al., 2018*). Venn diagrams were

created without prevalence or abundance cutoffs and include all low abundance taxa. Differentially abundant genera between transported and non-transported tadpoles were determined with the package 'ANCOMBC2' (version 2.2.2) (*Lin, 2023*).

We used 'Sourcetracker' (*Knights et al., 2011*) on unrarefied ASVs to determine proportions of the communities of adult caregivers ('source') that can be detected on tadpoles that were collected from their back and then reared in artificial cups for one month ('sink'). For this approach, we defined the communities of each caregiver as source and transported tadpoles as sinks and reported minimum, maximum and median source proportions detected among transported tadpoles.

ASVs shared between the transported tadpole and its transporting caregiver were identified using the function 'common_taxa' of the package 'phylosmith' (*Smith, 2023*). We then evaluated the presence and relative abundance of these shared taxa in transported and non-transported tadpoles, in their respective aquatic cup environments, and in the water used to refill their cups. We converted relative abundances to binary presence-absence information for illustration in a bubble chart and determined the ten taxa with the highest relative abundance for each transporting caregiver individually from the unrarefied dataset.

## Quantitative analysis of 16S rRNA gene copy numbers with digital PCR (dPCR)

Absolute abundances were estimated for ASVs that were shared between tadpoles after a one-month growth period and their respective caregivers, and for which Sourcetracker analysis identified the caregiver as a likely source of microbiota. We followed the quantitative sequencing framework described by *Barlow et al., 2020*, measuring total microbial load via digital PCR (dPCR) with the same universal 16 S rRNA primers used to amplify the V4 region in our sequencing dataset. Absolute 16 S rRNA gene copy numbers obtained from dPCR were then multiplied by the relative abundances from our amplicon sequencing dataset to calculate ASV-specific scaled absolute abundances. All dPCR reactions were carried out on a QIAcuity Digital PCR System (Qiagen) using Nanoplates with a 8.5 K partition configuration, using the following cycling program: 95°C for 2 min, 40 cycles of 95°C for 30 s and 52°C for 30 s and 72°C for 1 min, followed by 1 cycle of 40°C for 5 min. Reactions were prepared using the QIAcuity EvaGreen PCR Kit (Qiagen, Cat. No. 250111) with 2 µL of DNA template per reaction, following the manufacturer's protocol, and included a negative no-template control and a cleaned and sequenced PCR product as positive control. Samples were measured in triplicates and serial dilutions were performed to ensure accurate quantification. Data were processed with the QIAcuity Software Suite (v3.1.0.0). The threshold was set based on the negative and positive controls in 1D scatterplots. We report mean copy numbers per microliter with standard deviations, correcting for template input, dPCR reaction volume, and dilution factor. Mean gene copy numbers per tadpole were additionally calculated by accounting for the DNA extraction (elution) volume.

## Acknowledgements

We would like to thank Ami S Bhatt for her support in facilitating the contributions of Mai Dvorak to this work. We are grateful to the staff of the CNRS Guyane, the Nouragues Ecological Research Station and the ONF, especially Patrick Chatelet and Jennifer Devillechabrol, for logistic and moral support in the field. We thank Daniel Shaykevich for reviewing all versions of the manuscript, Camilo Rodríguez-Lopez for statistical consultation and Philippe Gaucher for endless knowledge, discussions and guidance in the world of amphibians. This research was conducted at Stanford University, which is located on the ancestral and unceded land of the Muwekma Ohlone tribe. Our field work was conducted in the Nature Reserve 'Les Nouragues' in French Guiana, that was founded on land ancestrally inhabited by the Amerindien Nouragues ('Norak') tribe.

## Additional information

### Competing interests
Lauren A O'Connell: Reviewing editor, eLife. The other authors declare that no competing interests exist.

## Funding

| Funder | Grant reference number | Author |
|---|---|---|
| National Institutes of Health | DP2HD102042 | Lauren A O'Connell |
| McKnight Foundation | | Lauren A O'Connell |
| Austrian Science Fund | Erwin Schroedinger Stipend J-4526B | Marie-Therese Fischer |
| Good Ventures Foundation | | David A Relman |

The funders had no role in study design, data collection and interpretation, or the decision to submit the work for publication.

## Author contributions

Marie-Therese Fischer, Conceptualization, Data curation, Formal analysis, Supervision, Investigation, Visualization, Methodology, Writing – original draft, Project administration, Writing – review and editing; Katherine S Xue, Elizabeth K Costello, Data curation, Supervision, Writing – review and editing; Mai Dvorak, Data curation, Formal analysis, Validation, Methodology, Writing – original draft, Writing – review and editing; Gaelle Raboisson, Anna Robaczewska, Data curation; Stephanie N Caty, Conceptualization, Supervision, Writing – review and editing; David A Relman, Resources, Supervision, Funding acquisition, Writing – review and editing; Lauren A O'Connell, Resources, Funding acquisition, Writing – review and editing

## Author ORCIDs

Marie-Therese Fischer ⦿ https://orcid.org/0000-0002-6693-8662
Katherine S Xue ⦿ https://orcid.org/0000-0002-4094-3615
Elizabeth K Costello ⦿ https://orcid.org/0000-0002-6441-2931
Mai Dvorak ⦿ https://orcid.org/0000-0002-8481-7063
Stephanie N Caty ⦿ https://orcid.org/0000-0002-9931-4944
David A Relman ⦿ https://orcid.org/0000-0001-8331-1354
Lauren A O'Connell ⦿ https://orcid.org/0000-0002-2706-4077

## Ethics

Approval of the field monitoring by the scientific institution of the authors:The monitoring project in the Nouragues was reviewed by the Administrative Panel on Laboratory Animal Care (APLAC) at Stanford University and approved under the APLAC-protocol 33691 (Protocol Director LAO). The APLAC committee is Stanford's Institutional Animal Care and Use Committee (IACUC). It is appointed by the University Vice Provost and Dean of Research. Approval from local field site: Non-invasive field experiments included in this publication were approved by the scientific committee of the Nouragues Reserve. The field station in the Nouragues Reserve is managed by the CNRS (Centre National de la Recherche Scientifique), is part of the USR 3456 LEEISA (Laboratoire Ecologie, Evolution Interactions des Systèmes Amazoniens, founded in 2016) and located in the nature reserve 'Les Nouragues', which is operated by the Office national des forêts (ONF) Guyane. Additional tissue collections were approved by the Direction Générale des Territoires et de la Mer (DGTM approval number # R03-2022-08-10-00001).

Reviewer #1 (Public review): https://doi.org/10.7554/eLife.103331.3.sa1
Reviewer #2 (Public review): https://doi.org/10.7554/eLife.103331.3.sa2
Author response https://doi.org/10.7554/eLife.103331.3.sa3

# Additional files

## Supplementary files
MDAR checklist

Supplementary file 1. Supplementary tables supporting the main findings of this study. (**a**) 16S rRNA gene copy number variations in embryos and jelly measured by ddPCR. (**b**) Source proportions of Os and Rv communities in the microbiome of tadpoles transported by Os, and their sibling

transported by Rv or non-transported. (**c**) Taxonomic composition of microbiome samples from wild tadpoles and adults. (**d**) Statistical analysis of alpha diversity measures across species and life stages and environments. (**e**) PERMANOVA results for Principal Coordinate-located in the nature reserve 'Les Analysis on Bray Curtis distances. (**f**) Identity and number of core taxa across abundance and prevalence cutoffs. (**h–j**) Shared ASVs between transported tadpoles and their caregivers, including relative and scaled absolute abundances and presence in non-transported tadpoles.

Supplementary file 2. Differentially abundant genera between transported and non-transported tadpoles. Differentially abundant genera between experimental conditions (N = 10 transported, N = 8 non-transported tadpoles) were determined with the R package ANCOMBC2.

### Data availability

The dataset titled "Effects of parental care on skin microbial community composition in poison frogs" has been reviewed and approved for publication in Dryad. It is publicly accessible via the link: https://doi.org/10.5061/dryad.s4mw6m9g9. Related software files are published and publicly available on Zenodo: https://doi.org/10.5281/zenodo.13766778.

The following datasets were generated:

| Author(s) | Year | Dataset title | Dataset URL | Database and Identifier |
|---|---|---|---|---|
| Fischer M, Xue K, Costello E, Dvorak M, Raboisson G, Robaczewska A, Caty S, Relman D, O'Connell L | 2024 | Effects of parental care on skin microbial community composition in poison frogs | https://doi.org/10.5061/dryad.s4mw6m9g9 | Dryad Digital Repository, 10.5061/dryad.s4mw6m9g9 |
| Fischer MT, Xue KS, Costello EK, Dvorak M, Raboisson G, Robaczewska A, Caty SN, Relman DA, O'Connell LA | 2024 | Effects of parental care on skin microbial community composition in poison frogs | https://doi.org/10.5281/zenodo.13766778 | Zenodo, 10.5281/zenodo.13766778 |

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
