## [Editor Report · eLife Assessment]

This study provides an **important** perspective on the influence of parental care in the establishment of the amphibian microbiome. Through a combination of cross-fostering experimental work, comparative analysis, and developmental time series, the authors provide **compelling** evidence that vertical transmission through care is possible, and **solid** but somewhat preliminary evidence that it plays a significant role in shaping frog skin microbiomes in nature or across time. This work will be of interest to researchers studying the evolution of parental care and microbiomes in vertebrates.

---

## [Referee Report · Reviewer #1 (Public review)]

Summary:

This manuscript describes a series of lab and field experiments to understand the role of tadpole transport in shaping the microbiome of poison frogs in early life. The authors conducted a cross-foster experiment in which *R. variabilis* tadpoles were carried by adults of their own species, carried by adults of another frog species, or not carried at all. After being carried for 6 hours, tadpole microbiomes resembled those of their caregiving species. Next, the authors reported higher microbiome diversity in tadpoles of two species that engage in transport-based parental care compared to one species that does not. Finally, they collected tadpoles either from the backs of an adult (i.e., they had recently been transported) or from eggs (i.e., not transported) but did not find significant overlap in microbiome composition between transported tadpoles and their parents.

Strengths:

The cross-foster experiment and the field experiment that reared transported and non-transported tadpoles are creative ways to address an important question in animal microbiome research. Together, they imply a small role for parental care in the development of the tadpole microbiome. The manuscript is generally well-written and easy to understand. The authors make an effort (improved since the first version of the manuscript) to acknowledge the limitations of their experimental design.

Weaknesses:

Cross-foster experiment: The "heterospecific transport" tadpoles were manually brushed onto the back of the surrogate frog, while the "biological transport" tadpoles were picked up naturally by the parent. It is challenging to interpret the effect of caregiver species since it is conflated with the method of attachment to the parent.

Cross-species analysis: The authors attribute the difference in diversity to parental care behavior, but the comparison only includes n=2 transporting species and n=1 non-transporting species that differ in many other ways. I would also add that increased diversity is not necessarily an expectation of vertical transmission. Similarity between adults and tadpoles is likely a more relevant outcome for vertical transmission, but the authors did not find any evidence that tadpole-adult similarity was any higher in species with tadpole transport. In fact, tadpoles and adults were more similar in the non-transporting species than in one of the transporting species (lines 296-298), which seems to directly contradict the authors' hypothesis.

---

## [Referee Report · Reviewer #2 (Public review)]

Summary:

Here, the Fischer et al. attempt to understand the role of parental care, specifically the transport of offspring, in the development of the amphibian microbiome. The amphibian microbiome is an important study system due to its association with host health and disease outcomes. This study provides vertical transfer of bacteria through parental transport of tadpoles as one mechanism, among others, influencing tadpole microbiome composition. This paper gives insight into the relative roles of the environment, species, and parental care in amphibian microbiome assembly.

The authors determine the time of bacterial colonization during tadpole development using PCR, observing that tadpoles were not colonized by bacteria prior to hatching from the vitelline membrane. This is an important finding for amphibian microbiome research and I would be curious to see if this is seen broadly across amphibian species. By doing this, the impact of transport can be more accurately assessed in their laboratory experiments. The authors found that caregiver species influenced community composition, with transported tadpoles sharing a greater proportion of their skin communities with the transporting species.

In a comparison of three sympatric amphibian species that vary in their reproductive strategies, the authors found that tadpole community diversity was not reflective of habitat diversity, but may be associated with the different reproductive strategies of each species. Parental care explained some of the variance of tadpole microbiomes between species, however, transportation by conspecific adults did not lead to more similar microbiomes between tadpoles and adults compared to species that do not exhibit parental transport. This finding is in agreement with the understanding that the amphibian microbiome is distinct between developmental stages (eggs/tadpoles/adults) and also that amphibian microbiome composition is generally species specific.

When investigating contributions of caretakers to transported offspring, the authors found that tadpole-adult pairs with a history of direct contact were not more similar than tadpole-adult pairs lacking that history. This conclusion was surprising when considering the direct contact between the adults and tadpoles, however if only certain taxa from the adults are capable of colonizing tadpoles, then one could expect that similar ASVs might be donated between tadpole-adult pairs.

I did not find any major weaknesses in my review of this paper. I think that the data and conclusions here are of value to other researchers looking into the assembly of the amphibian microbiome. This paper offers insight into how tadpole-transport could influence the microbiome and adds to our overall understanding of amphibian microbiome assembly across the varied life histories of frogs.

---

## [Author Response]

The following is the authors’ response to the original reviews

**Public Reviews:**

**Reviewer #1:**
(1) Developmental time series:It was not entirely clear how this experiment relates to the rest of the manuscript, as it does not compare any effects of transport within or across species.

Implemented Changes:

The importance of species arrival timing for community assembly is addressed in both the introduction and discussion. To accommodate the reviewer’s concerns and further emphasize this point, we have added a clarifying sentence to the results section and included an illustrative example with supporting literature in the discussion.

Results: Clarifying the timing of initial microbial colonization is essential for determining whether and how priority effects mediate community assembly of vertically transmitted microbes in early life, or whether these microbes arrive into an already established microbial landscape. We used non-sterile frogs of our captive laboratory colony (…)

Discussion: For example, early microbial inoculation has been shown to increase the relative abundance of beneficial taxa such as *Janthinobacterium lividum* (Jones et al., 2024), whereas efforts to introduce the same probiotic into established adult communities have not led to long-term persistence (Bletz, 2013; Woodhams et al., 2016).

(2) Cross-foster experiment:The "heterospecific transport" tadpoles were manually brushed onto the back of the surrogate frog, while the "biological transport" tadpoles were picked up naturally by the parent. It is a little challenging to interpret the effect of caregiver species since it is conflated with the method of attachment to the parent. I noticed that the uptake of Os-associated microbes by Os-transported tadpoles seemed to be higher than the uptake of Rv-associated microbes by Rv-associated tadpoles (comparing the second box from the left to the rightmost boxplot in panel S2C). Perhaps this could be a technical artifact if manual attachment to Os frogs was more efficient than natural attachment to Rv frogs.I was also surprised to see so much of the tadpole microbiome attributed to Os in tadpoles that were not transported by Os frogs (25-50% in many cases). It suggests that SourceTracker may not be effectively classifying the taxa.

Implemented Changes:

Methods (Study species, reproductive strategies and life history): *Oophaga sylvatica* (*Os*) (Funkhouser, 1956; CITES Appendix II, IUCN Conservation status: Near Threatened) is a large, diurnal poison frog (family *Dendrobatidae*) inhabiting lowland and submontane rainforests in Colombia and Ecuador. While male *Os* care for the clutch of up to seven eggs, females transport 1-2 tadpoles at a time to water-filled leaf axils where tadpoles complete their development (Pašukonis et al., 2022; Silverstone, 1973; Summers, 1992). Notably, females return regularly to these deposition sites to provision their offspring with unfertilized eggs.

Discussion: Most poison frogs transport tadpoles on their backs, but the mechanism of adherence remains unclear. Similar to natural conditions, tadpoles that are experimentally placed onto a caregiver’s back also gradually adhere to the dorsal skin, where they remain firmly attached for several hours as the adult navigates dense terrain. Although transport durations were standardized, species-specific factors- such as microbial density at the contact site, microbial taxa identity, and skin physiology such as moisture -could influence microbial transmission between the transporting frog and the tadpole. While these differences may have contributed to varying transmission efficacies observed between the two frog species in our experiment, none of these factors should compromise the correct microbial source assignment. We thus conclude that transporting frogs serve as a source of microbiota for transported tadpoles. However, further studies on species-specific physiological traits and adherence mechanisms are needed to clarify what modulates the efficacy of microbial transmission during transport, both under experimental and natural conditions.

Methods (Vertical transmission): Cross-fostering tadpoles onto non-parental frogs has been used previously to study navigation in poison frogs (Pašukonis et al., 2017). According to our experience, successful adherence to both parent and heterospecific frogs depends on the developmental readiness of tadpoles, which must have retracted their gills and be capable of hatching from the vitelline envelope through vigorous movement. Another factor influencing cross-fostering success is the docility of the frog during initial attachment, as erratic movements easily dislodge tadpoles before adherence is established. *Rv* are small, jumpy frogs that are easily stressed by handling, making experimental fostering of tadpoles—even their own— impractical. Therefore, we favored an experimental design where tadpoles initiate natural transport and parental frogs pick them up with a 100% success rate. We chose the poison frog *Os* as foster frogs because adults are docile, parental care in this species involves transporting tadpoles, and skin microbial communities differ from *Rv*- a critical prerequisite for our SourceTracker analysis. The use of the docile *Os* as the foster species enabled a 100% cross-fostering success rate, with no notable differences in adherence strength after six hours.

Methods (Sourcetracker Analysis): To assess training quality, we evaluated model selfassignment using source samples. We selected the model trained on a dataset rarefied to the read depth of the adult frog sample with the lowest read count (48162 reads), as it showed the best overall self-assignment performance, whereas models trained on datasets rarefied to the lowest overall read depth performed worse. Unlike studies using technical replicates, our source samples represent distinct biological individuals and sampling timepoints, where natural microbiome variability is expected within each source category. Consequently, we considered self-assignment rates above 70% acceptable. All source samples were correctly assigned to their respective categories (*Rv*, *Os*, or control), but with varying proportions of reads assigned as 'Unknown'. Adult frog sources were reliably selfidentified with high confidence (*Os*: 97.2% median, IQR = 1.4; *Rv*: 76.3% median, IQR = 38.1). Adult *R. variabilis* frogs displayed a higher proportion of 'Unknown' assignments compared to *O. sylvatica*, likely reflecting greater biological variability among individuals and/or a higher proportion of rare taxa not well captured in the training set. The control tadpole source showed lower self-assignment accuracy (median = 30.5%, IQR = 17.1), as expected given the low microbial biomass of these samples, which resulted in low read depth. Low readdepth limits the information available to inform the iterative updating steps in Gibbs sampling and reduces confidence in source assignments. We therefore verified the robustness of our results by performing the second Sourcetracker analysis as described above, training the model only on adult sources and assigning all tadpoles, including lowbiomass controls, as sinks (as described above). Self-assignment rates for the second training set varied (*O. sylvatica*: 79.2% median, IQR = 29; *R. variabilis*: 96.6% median, IQR = 3.7), while results remained consistent across analyses, supporting the reliability of our findings.

(3) Cross-species analysis:Like the developmental time series, this analysis doesn't really address the central question of the manuscript. I don't think it is fair for the authors to attribute the difference in diversity to parental care behavior, since the comparison only includes n=2 transporting species and n=1 non-transporting species that differ in many other ways. I would also add that increased diversity is not necessarily an expectation of vertical transmission. The similarity between adults and tadpoles is likely a more relevant outcome for vertical transmission, but the authors did not find any evidence that tadpole-adult similarity was any higher in species with tadpole transport. In fact, tadpoles and adults were more similar in the non-transporting species than in one of the transporting species (lines 296-298), which seems to directly contradict the authors' hypothesis. I don't see this result explained or addressed in the Discussion.

To address the reviewer’s concerns, we implemented the following changes:

Results:

We rephrased the following sentence from the results part:

“These variations may therefore be linked to differing reproductive traits: *Af* and *Rv* lay terrestrial egg clutches and transport hatchlings to water, whereas *Ll*, a non-transporting species, lays eggs directly in water.”

To read

“These variations may therefore reflect differences in life history traits among the three species.”

We moved the information on differing reproductive strategies into the Discussion, where it contributes to a broader context alongside other life history traits that may influence community diversity.

Discussion (1): We added to our discussion that increased microbial diversity was not an expected outcome of vertical transmission.

“However, increased microbial diversity is not a known outcome of vertical transmission, and further studies across a broader range of transporting and non-transporting species are needed to assess the role of transport in shaping diversity of tadpole-associated microbial communities.”

Discussion (2): Likewise, communities associated with adults and tadpoles of transporting species were no more similar than those of non-transporting species. While poison frog tadpoles do acquire caregiver-specific microbes during transport, most of these microbes do not persist on the tadpoles' skin long-term. This pattern can likely be attributed to the capacity of tadpole skin- and gut microbiota to flexibly adapt to environmental changes (Emerson & Woodley, 2024; Santos et al., 2023; Scarberry et al., 2024). It may also reflect the limited compatibility of skin microbiota from terrestrial adults with aquatic habitats or tadpole skin, which differs structurally from that of adults (Faszewski et al., 2008). As a result, many transmitted microbes are probably outcompeted by microbial taxa continuously supplied by the aquatic environment. Interestingly, microbial communities of the non-transporting *Ll* were more similar to their adult counterparts than those of poison frogs. This pattern might reflect differences in life history among the species. While adult *Ll* commonly inhabit the rock pools where their tadpoles develop, adults of the two poison frog species visit tadpole nurseries only sporadically for deposition. These differences in habitat use may result in adult *Ll* hosting skin microbiota that are better adapted to aquatic environments as compared to *Rv* and *Af*. Additionally, their presence in the tadpoles’ habitat could make *Ll* a more consistent source of microbiota for developing tadpoles.

(4) Field experiment: The rationale and interpretation of the genus-level network are not clear, and the figure is not legible. What does it mean to "visualize the microbial interconnectedness" or to be a "central part of the community"? The previous sentences in this paragraph (lines 337-343) seem to imply that transfer is parent-specific, but the genuslevel network is based on the current adult frogs, not the previous generation of parents that transported them. So it is not clear that the distribution or co-distribution of these taxa provides any insight into vertical transmission dynamics.

Implemented Changes:

We appreciate the reviewer’s close reading and understand how the inclusion of the network visualization without further clarification may have led to confusion. To clarify, the network was constructed from all adult frogs in the population, including—but not limited to—the parental frogs examined in the field experiment. We do not make any claims about the origin of the microbial taxa found on parental frogs. Rather, our aim was to illustrate how genera retained on tadpoles (following potential vertical transmission) contribute to the skin microbial communities of adult frogs of this population beyond just the parental individuals. This finding supports the observation that these retained taxa are generally among the most abundant in adult frogs. However, since this information is already presented in Table S8 and the figure is not essential to the main conclusions, we have removed Supplementary Figure S5 and the accompanying sentence: “A genus-level network constructed from 44 adult frogs shows that the retained genera make up a central part of the community of adult *Rv* in wild populations (Fig. S5).” We have adjusted the Methods section accordingly.

**Reviewer #2:**
I did not find any major weaknesses in my review of this paper. The work here could potentially benefit from absolute abundance levels for shared ASVs between adults and tadpoles to more thoroughly understand the influences of vertical transmission that might be masked by relative abundance counts. This would only be a minor improvement as I think the conclusions from this work would likely remain the same, however.

In response to the reviewer’s suggestion, we estimated the absolute abundance of specific ASVs for all samples of tadpoles in which Sourcetracker identified shared ASVs between adults and tadpoles. The resulting scaled absolute abundance values (in copies/μL and copies per tadpole) are provided in Table S10, and a description of the method has been incorporated into the revised Methods section of the manuscript. To support the robustness of this approach in our dataset, we additionally designed an ASV-specific system for ASV24902-*Methylocella*. Candidate primers were assessed for specificity by performing local BLASTn alignments against the full set of ASV sequences identified in the respective microbial communities of tadpoles. We optimized the annealing temperature via gradient PCR and confirmed primer specificity through Sanger sequencing of the PCR product (Forward: 5′–GAGCACGTAGGCGGATCT–3′ Reverse: 5′–GGACTACNVGGGTWTCTAAT–3′). Using this approach, we confirmed that the relative abundance of ASV24902 (18.05% in the amplicon sequencing data) closely matched its proportion of the absolute 16S rRNA copy number in transported tadpole 6 (18.01%). While we intended to quantify all shared ASVs, we were limited to this single target due to insufficient material for optimizing the assays. As this particular ASV was also detected in the water associated with the same tadpole, we chose not to include this confirmation in the manuscript. Nevertheless, the close match supports the reliability of our approach for scaling absolute abundances in this dataset.

Results: Absolute abundances of shared ASVs likely originating from the parental source pool (as identified by Sourcetracker) after one month of growth ranged from 7804 to 172326 copies per tadpole (Table S10).

Methods: Quantitative analysis of 16S rRNA copy numbers with digital PCR (dPCR)

Absolute abundances were estimated for ASVs that were shared between tadpoles after a one-month growth period and their respective caregivers, and for which Sourcetracker analysis identified the caregiver as a likely source of microbiota. We followed the quantitative sequencing framework described by Barlow et al. (2020), measuring total microbial load via digital PCR (dPCR) with the same universal 16S rRNA primers used to amplify the v4 region in our sequencing dataset. Absolute 16S rRNA copy numbers obtained from dPCR were then multiplied by the relative abundances from our amplicon sequencing dataset to calculate ASV-specific scaled absolute abundances. All dPCR reactions were carried out on a QIAcuity Digital PCR System (Qiagen) using Nanoplates with a 8.5K partition configuration, using the following cycling program: 95°C for 2 minutes, 40 cycles of 95°C for 30 seconds and 52°C for 30 seconds and 72°C for 1 minute, followed by 1 cycle of 40°C for 5 minutes. Reactions were prepared using the QIAcuity EvaGreen PCR Kit (Qiagen, Cat. No. 250111) with 2 µL of DNA template per reaction, following the manufacturer's protocol, and included a negative no-template control and a cleaned and sequenced PCR product as positive control. Samples were measured in triplicates and serial dilutions were performed to ensure accurate quantification. Data were processed with the QIAcuity Software Suite (v3.1.0.0). The threshold was set based on the negative and positive controls in 1D scatterplots. We report mean copy numbers per microliter with standard deviations, correcting for template input, dPCR reaction volume, and dilution factor. Mean copy numbers per tadpole were additionally calculated by accounting for the DNA extraction (elution) volume.

**Recommendations for the authors:**

**Reviewer #1:**
(1) Figure 1b summarizes the ddPCR data as a binary (detected/not detected), but this contradicts the main text associated with this figure, which describes bacteria as present, albeit in low abundances, in unhatched embryos (lines 145-147). Could the authors keep the diagram of tadpole development, which I find very useful, but add the ddPCR data from Figure S1c instead of simply binarizing it as present/absent?

We appreciate the reviewer’s positive feedback on the clarity of the figure. We agree that presenting the ddPCR data in a more quantitative manner provides a more accurate representation of bacterial abundance across developmental stages. In response, we have retained the developmental diagram, as suggested, and replaced the binary (detected/not detected) information in Figure 1B with rounded mean values for each stage. To complement this, we have included mean values and standard deviations in Table S1. The corresponding text in the main manuscript and legends has been revised accordingly to reflect these changes.

(2) More information about the foster species, *Oophaga sylvatica*, would be helpful. Are they sympatric with Rv? Is their transporting behavior similar to that of Rv?

We thank the reviewer for this helpful comment. In response, we have added further details on the biology and parental care behavior of *Oophaga sylvatica*, including information on its distribution range. The species does not overlap with *Ranitomeya variabilis* at the specific study site where the field work was conducted, although the species are sympatric in other countries. These additions have been incorporated into the Methods section under "Study species, reproductive strategies, and life history."

(3) Plotting the proportion of each tadpole microbiome attributed to *R. variabilis* and the proportion attributed to *O. sylvatica* on the same plot is confusing, as these points are nonindependent and there is no way for the reader to figure out which points originated from the same tadpole. I would suggest replacing Figure 1D with Figure S2C, which (if I understand correctly) displays the same data, but is separated according to source.

We agree with the reviewer that Figure S2C allows for clearer interpretation of our results. In response, we implemented the suggested change and replaced Figure 1D with the alternative visualization previously shown in Figure S2C, which displays the same data separated by source. To provide readers with a complementary overview of the full dataset, we have retained the original combined plot in the supplementary material as Figure S2D.

(4) On the first read, I found the use of "transport" in the cross-fostering experiment confusing until I understood that they weren't being transported "to" anywhere in particular, just carried for 6 hours. A change of phrasing might help readers here.

We acknowledge the reviewer’s concern and have replaced “transported” with “carried” to avoid confusion for readers who may be unfamiliar with the behavioral terminology. However, because “transport” is the term widely used by specialists to describe this behavior, we now introduce it in the context of the experimental design with the following phrasing:

“For this design, sequence-based surveys of amplified 16S rRNA genes were used to assess the composition of skin-associated microbial communities on tadpoles and their adult caregivers (i.e., the frogs carrying the tadpoles, typically referred to as ‘transporting’ frogs).”

(5) "Horizontal transfer" typically refers to bacteria acquired from other hosts, not environmental source pools (line 394).

We addressed this concern by rephrasing the sentence in the Discussion to avoid potential confusion. The revised text now reads:

“Across species, newborns might acquire bacteria not only through transfer from environmental source pools and other hosts (…)”

(6) The authors suggest that tadpole transport may have evolved in Rv and Af to promote microbial diversity because "increased microbial diversity is linked to better health outcomes" (lines 477-479). It is often tempting to assume that more diversity is always better/more adaptive, but this is not universally true. The fact that the Ll frogs seem to be doing fine in the same environment despite their lower microbiome diversity suggests that this interpretation might be too far of a reach based on the data here.

We appreciate the reviewer’s concern, agree that increased microbial diversity is not inherently advantageous and have revised the paragraph to make this clearer.

“While increased microbial diversity is not inherently advantageous, it has been associated with beneficial outcomes such as improved immune function, lower disease risk, and enhanced fitness in multiple other vertebrate systems.”

However, rather than claiming that greater diversity is always advantageous, we suggest that this possibility should not be excluded and consider it a relevant aspect of a comprehensive discussion. We also note that whether poison frog tadpoles perform equally well with lower microbial diversity remains an open question. Drawing such conclusions would require experimental validation and cannot be inferred from comparisons with an evolutionarily distant species that differs in life history.

**Reviewer #2:**
(1) Figure 2: Are the data points in C a subset (just the tadpoles for each species) of B? The numbers look a little different between them. The number of observed ASVs in panel B for Rv look a bit higher than the observed ASVs in panel C.

The data shown in panel C are indeed a subset of the samples presented in panel B, focusing specifically on tadpoles of each species. The slight differences in the number of observed ASVs between panels result from differences in rarefaction depth between comparisons: due to variation in sequencing depth across species and life stages, we performed rarefaction separately for each comparison in order to retain the highest number of taxa while ensuring comparability within each group. Although we acknowledge that this is not a standard approach, we found that results were consistent when rarefying across the full dataset, but chose the presented approach to better accommodate variation in our sample structure. This methodological detail is described in the Methods section:

“All alpha diversity analyses were conducted with datasets rarefied to 90% of the read number of the sample with the fewest reads in each comparison and visualized with boxplots.”

It is also noted in the figure legend: “The dataset was separately rarefied to the lowest read depth f each comparison.” We hope this clarification adequately addresses the reviewer’s concern and therefore have not made additional changes.

(2) Lines 304-305: in the Figure 4B plot, there appear to be 12 transported tadpoles and 8 non-transported tadpoles.

Thank you for catching this. We have corrected the plot and the associated statistics (alpha and beta diversity) in the results section as well as in the figure. Importantly, the correction did not affect any other results, and the overall findings and interpretations remain unchanged.

(3) Line 311: I think this should be Figure 4B.(4) Line 430: tadpole transport.(5) Line 431: I believe commas need to surround this phrase "which range from a few hours to several days depending on the species (Lötters et al., 2007; McDiarmid & Altig, 1999; Pašukonis et al., 2019)".

We thank the reviewer for the thorough review and have corrected all typographical and formatting errors noted in comments (3) – (5).